# Organization at criticality enables processing of time-varying signals by receptor networks

Angel Stanoev, Akhilesh P Nandan & Aneta Koseska[*] 

## Abstract

How cells utilize surface receptors for chemoreception is a recurrent question spanning between physics and biology over the past few decades. However, the dynamical mechanism for processing time-varying signals is still unclear. Using dynamical systems formalism to describe criticality in non-equilibrium systems, we propose generic principle for temporal information processing through phase space trajectories using dynamic transient memory. In contrast to short-term memory, dynamic memory generated via "ghost" attractor enables signal integration depending on stimulus history and thereby uniquely promotes integrating and interpreting complex temporal growth factor signals. We argue that this is a generic feature of receptor networks, the first layer of the cell that senses the changing environment. Using the experimentally established epidermal growth factor sensing system, we propose how recycling could provide self-organized maintenance of the critical receptor concentration at the plasma membrane through a simple, fluctuation-sensing mechanism. Processing of non-stationary signals, a feature previously attributed only to neural networks, thus uniquely emerges for receptor networks organized at criticality.

**Keywords** criticality; saddle-node bifurcation; transient memory
**Subject Categories** Computational Biology; Signal Transduction
**Mol Syst Biol.** (2020) 16: e8870

## Introduction

In a wide variety of biological processes including embryogenesis, immune cells motility, wound healing or cancer metastasis, cells sense and interpret time-varying chemical signals that reflect the non-stationary environment to which they readily adapt. It has been, for example, demonstrated that time-varying growth factor signals not only trigger corresponding phenotypic output in cells, but a range of input frequencies can bias towards a specific function (i.e. differentiation), irrespective of growth factor identity (Ryu *et al*, 2015). Cells can also direct their motility through continuously changing patterns of chemical signals such as travelling waves of chemoattractants (Skoge *et al*, 2014), using memory of stimulus

history to integrate conflicting signals (Foxman *et al*, 1999; Welf *et al*, 2012). Generally, a transient memory of stimulus history is a main requirement for systems that process time-varying signals, as a means to integrate temporal dependencies inherent in the signal (Hochreiter & Schmidhuber, 1997; Maass *et al*, 2002).

How cells sense the growth factors from their environment has been extensively studied using equilibrium and non-equilibrium descriptions of sensing through ligand binding/unbinding dynamics for stationary levels of receptors and ligands (Berg & Purcell, 1977; Bialek & Setayeshgar, 2005; Wang *et al*, 2007; Rappel & Levine, 2008; Endres & Wingreen, 2009; Mora & Wingreen, 2010). These studies provide analysis of the fundamental limits of ligand concentration sensing by direct mapping to receptor occupancy that serves as a proxy for receptor activity. However, these mapping properties cannot satisfy and thereby do not apply to systems where memory requirements are necessary for integrating time-varying signals.

Receptor activity dynamics, on the other hand, is not only influenced by ligand binding dynamics, but rather reflects the dynamics of the biochemical network in which the receptor is embedded (Stanoev *et al*, 2018). Non-trivial dynamical solutions can hereby emerge, in particular due to recurrent interactions between the network components (Reynolds *et al*, 2003; Tischer & Bastiaens, 2003). In a broad range of biological systems, positive feedback interactions give rise to bistable dynamics, which is considered to underlie memory features (Xiong & Ferrell, 2003; Wang *et al*, 2009; Burrill & Silver, 2010; Doncic *et al*, 2015). However, signal-induced switching between basal and high receptor activity states, and thereby permanent memory formation, limits response to upcoming stimuli (Stanoev *et al*, 2018). To overcome equivalent limitations of stable states, information processing in the context of real-time computations of sensory stimuli by neural microcircuits, universal frameworks using transient dynamics and state-dependent trajectories have been proposed (Maass *et al*, 2002; Durstewitz, 2003; Jaeger & Haas, 2004). These formalisms typically contain high-dimensional state representations and non-linear intrinsic activation dynamics of the neuron components (Maass *et al*, 2002; Jaeger & Haas, 2004; Ozturk & Principe, 2005) and thus cannot be directly translated to biochemical networks. Therefore, a conceptual framework that describes processing of time-varying signals on the level of cellular sensing networks is lacking.

We propose here a saddle-node (*SN*) "ghost" as a minimal dynamical mechanism that enables processing of time-varying

---

Department of Systemic Cell Biology, Max Planck Institute for Molecular Physiology, Dortmund, Germany
*Corresponding author. Tel: +49 2311332225; E-mail: aneta.koseska@mpi-dortmund.mpg.de

growth factor signals. Critical organization in a vicinity of a *SN* bifurcation enables transient memory of receptor activity to be realized via the metastable "ghost" state. In contrast to the short- or long-term memory that stem from stable attractors, we demonstrate that this transient memory is dynamic and thereby uniquely promotes integrating and interpreting complex temporal growth factor signals. A clear distinction between a transient memory that reflects a dynamical state to a kinetic relaxation of receptor activity in terms of the signal integration capabilities is also shown. Using single-molecule reaction–diffusion simulations on the other hand, we depict how such dynamic memory can be realized on molecular level. Based on the experimental findings that the epidermal growth factor receptor (EGFR) system operates at critical organization (Stanoev *et al*, 2018), we propose a fluctuation-sensing mechanism as a basis for self-organized maintenance at the critical region and discuss its limitations. We further discuss why organization at criticality represents a generic dynamical mechanism which enables processing of time-varying growth factor signals by cell surface receptors.

## Results

### Organization at criticality enables transient temporal memory of growth factor signals to be manifested in receptor activity

Systems that sense time-varying signals require memory in order to integrate the signal information (Hochreiter & Schmidhuber, 1997). A minimal cellular sensing network that accounts for memory in receptor activity ($R_a$) is a two-component toggle switch (Fig 1A), where the double-negative feedback (DNF) interaction (Reynolds *et al*, 2003; Tischer & Bastiaens, 2003; Stanoev *et al*, 2018) between the active receptor and an inactivating enzyme (e.g. a phosphatase), $P_{DNF,a}$, follows the law of mass action:

$$\frac{dR_a}{dt} = k_R\left[R_i(\alpha_1 R_i + \alpha_2 R_a + \alpha_3 LR_a) - \hat{\gamma}_{DNF}P_{DNF,a}R_a\right]$$
$$\frac{dP_{DNF,a}}{dt} = k_1\left[P_{DNF,i} - k_{2/1}P_{DNF,a} - \hat{\beta}_{DNF}P_{DNF,a}(R_a + LR_a)\right] \quad (1)$$

The system combines autocatalytic receptor activation and mutual inhibition mechanisms (Fig 1A) that govern the protein state transitions between the active ($R_a$, $P_{DNF,a}$) and inactive ($R_i$, $P_{DNF,i}$) states of the switch components. They are described in further detail in Materials and Methods with the corresponding parameters.

Bistability in receptor activity is exhibited between two saddle-node bifurcation points for a broad range of the bifurcation parameter—the $P_{DNF,T}/R_T$ concentration ratio $\propto \hat{\gamma}_{DNF}$—in the absence of stimulus input (Fig 1B, green shaded region). The two stable states correspond to the basal and the high receptor activity states. The system maintains bistability also for a certain range of inputs (Fig 1C, green shaded region). The effective input for the cells in this case is the fraction of ligand-bound receptors ($LR_a$) that reflects the extracellular ligand concentration (Materials and Methods). Since the temporal receptor activity dynamics upon changes in growth factor stimulation is governed by the receptor dose–response dynamics, it will also depend on the system's organization in parameter space (the $P_{DNF,T}/R_T$ concentration ratio). We therefore

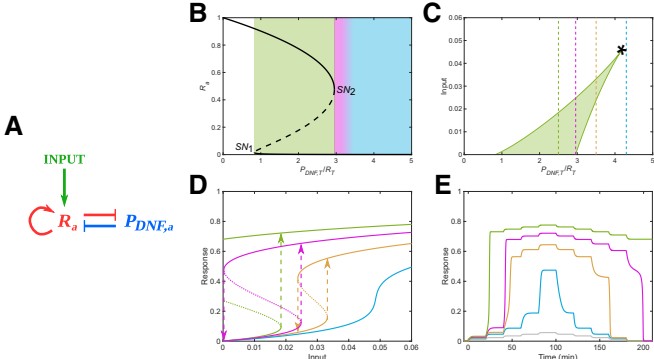

**Figure 1. Memory manifestation depending on parameter organization.**

A  Diagram of a two-component toggle switch between active receptors ($R_a$) and the deactivating enzyme, protein $P_{DNF,a}$. Input—fraction of ligand-bound receptors ($LR_a$). Molecular details described in Materials and Methods.

B  Bifurcation diagram of the $R$-$P_{DNF}$ toggle switch, depicting $R_a$ response with respect to $P_{DNF,T}/R_T$, in the absence of input. Shading: blue—monostable region, magenta—vicinity of the saddle-node (SN) bifurcation point and green—bistable region. Solid/dashed lines—stable/unstable steady states.

C  Two-parameter ($LR_a$, $P_{DNF,T}/R_T$) bifurcation diagram depicting the parameter space where bistability exists (green area). Vertical lines denote organization in irreversible bistable (green), critical (magenta), reversible bistable (yellow) and monostable (blue) organization.*—cusp bifurcation.

D  Steady-state receptor activity response for increasing input doses in the different organizations. Solid/dotted lines—stable/unstable steady states. Arrows—switch on/off points.

E  Temporal receptor activity responses to step-wise modulation of the input ($LR_a$, grey) for organization in the distinct parameter regimes.

Data information: In (D, E), the colours correspond to the respective system organization depicted with vertical lines in (C).

investigate next whether the stable attractor solutions could underlie emergence of transient memory in receptor activity.

For $P_{DNF,T}/R_T$ concentration ratio that corresponds to organization in the bistability region in the absence of growth factors (Fig 1C, green line), the numerical simulations show irreversible receptor activation. This is reflected both, in the receptor's steady-state response to changes of growth factor doses (Fig 1D, green) and in the temporal receptor activity profile (Fig 1E, green) upon step-wise modulation of growth factor input (Fig 1E, grey). The system thereby exhibits a temporal long-term memory to the presence of single growth factor stimulus: the receptor activity is irreversibly maintained at high levels after growth factor removal. In contrast, for system's organization in the reversible bistable regime (Fig 1C, yellow line), the receptor's activity response displays hysteresis with respect to the input doses that activate/deactivate it (Fig 1D, yellow). This induces a short-term memory only regarding the growth factor dose that activates the system, but the memory is not reflected in the temporal receptor activity profile (Fig 1E, yellow). Thus, the receptor activity is not prolonged in time upon removal of the growth factor input. Temporal response without prolonged receptor activity after input removal was also observed for $P_{DNF,T}/R_T$ concentration ratios that correspond to organization in the monostable regime (Fig 1C and E, blue lines). In this regime however, there is no hysteresis and thus no memory of the growth factor dose that activates the system (Fig 1D, blue). These results indicate that both

the long- and the short-term memory that result from the presence of stable attractors do not fulfil the conditions necessary for processing time-varying inputs. The long-term memory is not transient and thereby it will inhibit responsiveness to upcoming cues (Stanoev et al, 2018), whereas the short-term memory only corresponds to the growth factor dose that activates the system and is not reflected in the temporal receptor activity profile.

For organization in the vicinity of the saddle-node bifurcation point (Fig 1B and C, magenta), the numerical simulations demonstrate the presence of memory of the dose that activates the receptor (Fig 1D, magenta). The activation of the receptor, similarly as in the reversible and irreversible bistable regimes, occurs in a switch-like manner at a threshold input dose, indicating that spurious activation is filtered out. However, additionally, high receptor activity was transiently maintained over time after removal of the growth factor (Fig 1E, magenta). This shows that critical organization confers to the sensing system a transient memory of the previous input-driven activation.

## Saddle-node "ghost" as a dynamical mechanism of transient temporal memory

To understand how transient memory occurs for critical organization of the system in the vicinity of the saddle-node bifurcation point, and in particular how it is distinguished from short- and long-term memory, we studied qualitatively the dynamical $R_a$-$P_{DNF,a}$ behaviour. We analysed how the phase space trajectories evolve in relation to the changes in the geometry of the underlying phase space as a function of a pulsed stimulus. Generally, the relative positioning of the nullclines, which are determined by the system parameters, shapes the phase space geometry. In non-autonomous or input-driven systems, either the geometry of the underlying phase space can be altered (change in the positioning, shape and size of the attractors), or its topology (change in the number or stability of the attractors) (Verd et al, 2014; Jimenez et al, 2017). We also estimated the associated quasi-potential landscapes (Strogatz, 2018; Fig 2; Materials and Methods; and Verd et al, 2014) that depict the energy-like levels associated with the states. The phase space trajectories flow downhill the landscapes, towards the valleys defined by the stable steady states.

We first consider organization in the monostable regime (Fig 1B and C, blue), where the system does not exhibit any memory in receptor activity (Fig 1E). In this case, the pulsed stimulus induced changes in the phase space geometry of the system (adding/removing stimulus: $i \rightarrow ii/ii \rightarrow iii$, blue transitions, Fig 2A), thereby triggering continuous and reversible re-positioning of the single steady-state attractor that captivates the state trajectory. This leads to receptor response that closely follows the input (Fig 2A left and inset). However, when in the absence of stimulus, the system is poised in the valley of basal receptor activity in the double-well quasi-potential landscape characteristic for the bistable organization (Fig 1B and C, green), a topological phase space change where this state vanishes occurs at a threshold signal concentration. This results in a transition to the high receptor activity state (Fig 2B, $i \rightarrow ii$ green transition), which also explains the previously demonstrated switch-like response to increased growth factor doses (Fig 1D, green). Upon signal removal, the reverse topological change leads to re-establishing of bistability ($ii \rightarrow iii$ green transition). However, the trajectory remains in the occupied high activity

stable steady state (green circle in Fig 2B middle). Thus, the first pulse will activate the receptors and this will hinder further responsiveness to upcoming stimuli due to the long-term memory that results from this stable attractor organization (Fig 2B left, inset). In contrast, for organization in the reversible bistable regime, the changes in the topology of the phase space induced by the pulsed stimulus allow for reversible switching between the two stable attractors, the basal and high receptor activity (topological transitions are omitted from Fig 2 for clarity). These topological changes thereby also guide the phase space trajectory such that the time spent in the stable attractors is equivalent to the administration time of the stimuli, resulting in the absence of prolonged receptor activity upon growth factor removal (Fig 1E, yellow).

When the system is positioned in the vicinity of the saddle-node bifurcation point (Fig 1B and C, magenta), a supra-threshold input pulse induces transition from the basal monostable to the high activity monostable state (Fig 2C, $i \rightarrow ii$ magenta transition) via the bistable region in a switch-like manner (Fig 1D). Upon input removal, these consecutive topological transitions are reversed. However, there is a delay between establishing the single stable attractor (magenta state $iii$) and the system trajectory converging to it (magenta state $iv$), resulting in prolonged receptor activity before relaxation to basal level (Fig 2C left). This delay results in a transient memory of receptor activity that does not hinder further responsiveness of the system (Fig 2C inset).

The transient memory is a consequence of the critical dynamical behaviour near the *SN* bifurcation point. In this organization, the nullclines intersect only once, indicating a single stable steady state of basal receptor activity. However, they are positioned very close to one another in the phase space area where the steady state of high receptor activity is stable for bistable organization (compare Fig 2B and C, middle), resulting in a quasi-potential landscape with a very shallow slope (Fig 2C middle, inset). Thus, when the system transits back from the bistable to the monostable region, the remnant of the saddle node that disappeared in this transition generates a metastable state that continues to capture the incoming trajectories (see Movie EV1 for a stochastic simulation of this process). Such delayed dynamics, different from a slow relaxation kinetics, is referred to as a feature of a "ghost" attractor (Strogatz & Westervelt, 1989) and has been previously shown in some driven dynamical systems such as ferroelectrics or semiconductor lasers (Rogister et al, 2003). We have demonstrated here, however, that saddle-node "ghost" serves as a unique dynamical mechanism of transient temporal memory, enabling receptor activity to be maintained at high levels for a limited period of time after growth factor removal.

## Dynamic temporal memory is a prerequisite for processing time-varying growth factor signals

To integrate and interpret the information contained in time-varying growth factor profiles however, the transient temporal memory in receptor activity must be dynamic. In other words, the total duration of the receptor activity should depend on the previous stimulus history. We therefore probed the features of the transient memory resulting from the saddle-node "ghost" using a train of two subsequent growth factor pulses with inter-pulse interval shorter than the duration of the memory. The numerical simulations demonstrated that the receptor activity dynamics could

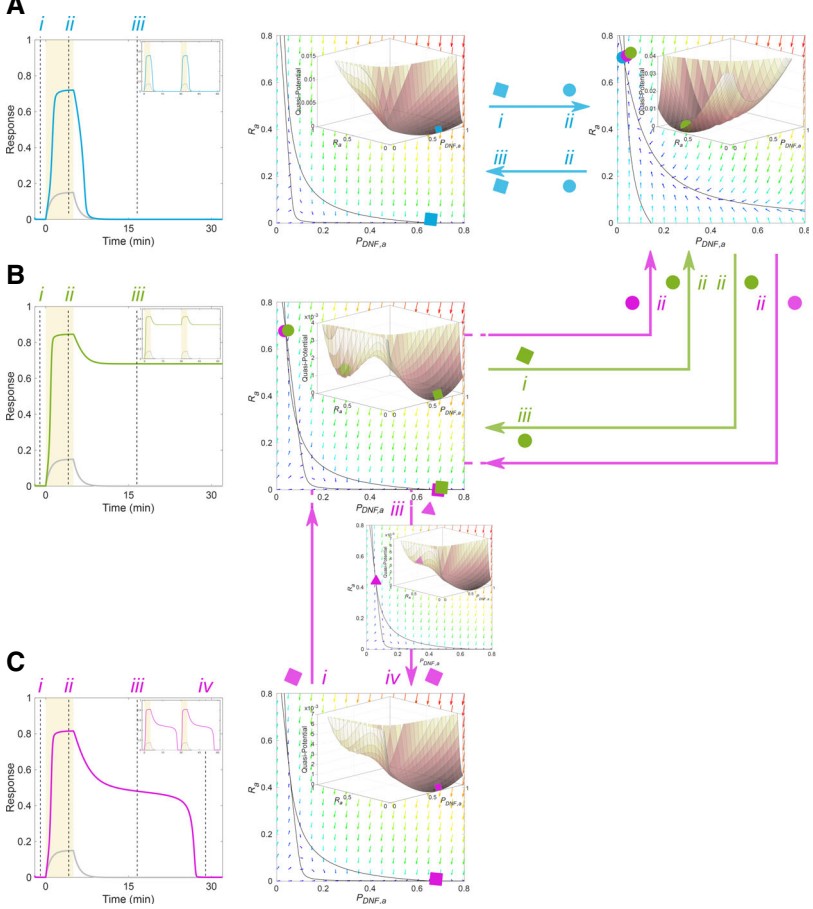

**Figure 2. Qualitative $R_a$-$P_{DNF,a}$ behaviour with respect to phase space changes to pulsed input.**

A   Left: receptor response (blue) to single growth factor pulse (yellow) and respective profile of ligand-bound receptors ($LR_a$, grey) for positioning in the monostable regime ($P_{DNF,T}/R_T$ = 4.3). Inset: responsiveness to subsequent input pulses. Middle/right: phase space diagram and nullclines intersecting at the basal/high activity receptor steady state denoted with blue squares/circles, respectively. Blue arrows: phase space transitions upon administering and removal of stimulus ($i \rightarrow ii$, $ii \rightarrow iii$, respective time points denoted in left plot). Insets: calculated quasi-potential landscapes.

B   Same as in (A), only for positioning in the bistable regime ($P_{DNF,T}/R_T$ = 2.5). Green arrows: phase space transitions ($i \rightarrow ii$, $ii \rightarrow iii$).

C   Same as in (A), for positioning at the critical transition between monostability and bistability ($P_{DNF,T}/R_T$ = 2.957). Magenta arrows: signal administration ($i \rightarrow ii$) and removal ($ii \rightarrow iii \rightarrow iv$). The $iii \rightarrow iv$ transition and the associated phase space plot demonstrate the existence of a "ghost" attractor. Parameters for (A–C) as in Fig 1.

rapidly adapt to the second growth factor pulse, and the period in which the receptor activity is maintained high is longer than in the case of a single pulse stimulation (compare Fig 3A, magenta to Fig 2C, left). In contrast, the time-frame in which the receptor activity was maintained high for the other temporal memory manifestation, organization in the irreversible bistable regime, was equivalent as for a single pulse (Fig 3A, green). The receptor activity profile in the absence of temporal memory on the other hand, such as for organization in the reversible bistable and monostable regimes (Fig 3A yellow and blue, respectively), closely followed that of the growth factor stimuli.

We next simulated 1,000 different non-periodic growth factor pulse trains by randomly distributing twelve 5-min growth factor pulses over period of 480 min. The total duration of high receptor activity, reflecting the degree of history-dependent signal integration, was estimated for the individual realizations. The resulting distributions strongly depended on the organization of the system in parameter space. In the reversible bistable and monostable regimes,

the distributions were narrow and closely reflected the total duration of growth factor pulses (Fig 3B, yellow and blue), whereas for irreversible bistable positioning, the receptor activity was constant over all realizations and equivalent to the integration time (Fig 3B, green). In contrast, for positioning in the critical vicinity of the *SN* bifurcation, the total duration of receptor activity was highly variable (Fig 3B, magenta), reflecting the varying degrees of history-dependent signal integration that depend on the temporal signature of the signal profile. This variability is also depicted by the exemplary temporal receptor activity profiles following different periodic growth factor pulse trains (Fig 3D).

For the different growth factor pulse trains, we also estimated the number of disjoint intervals of receptor activity over the integration time (black line segments in Fig 3D, middle), thereby reflecting whether the response follows the temporal partitioning of the signal. Again, in the presence of dynamic memory, the respective distribution was broad (Fig 3C, magenta), emphasizing the varying degrees of partitioning in response to complex signals. In the irreversible

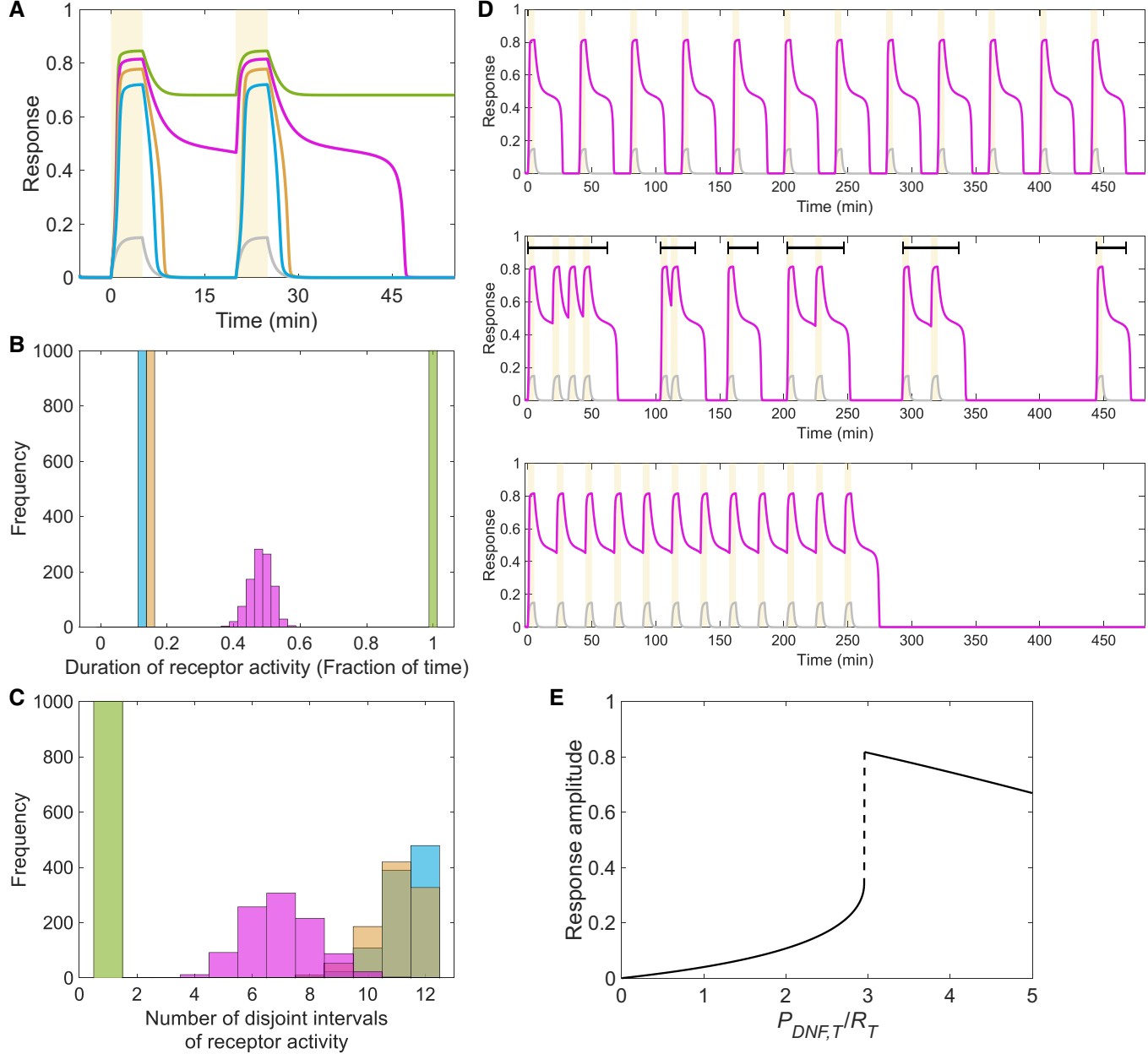

**Figure 3. Dynamic transient memory uniquely enables processing time-varying growth factor signals.**

A   Receptor responsiveness to two subsequent 5-minute growth factor pulses for different organizations of the system, denoted by colours as in Fig 1C. Yellow shaded area: growth factor pulse duration. Grey temporal profile: input ($LR_a$).

B   Distribution of total duration of receptor activity (as a fraction of total time, 480 min) calculated for growth factor pulse trains constructed from 12 subsequent 5-min pulses randomly distributed over time. Different colours denote responses for different system's organization (equivalent to Fig 1C). The distributions are generated from 1,000 independent realizations.

C   Distribution of number of disjoint intervals of receptor activity (top black lines in D), estimated for the growth factor pulse trains in (B).

D   Exemplary temporal receptor activity profiles for different growth factor pulse trains realizations for critical organization. Black line segments: disjoint intervals of receptor activity. Yellow shaded area: growth factor pulse duration. Grey temporal profile: input ($LR_a$).

E   Dynamic range of receptor activation for input that activates the system ($LR_a$ = 0.15), as a function of $\frac{P_{DNF,T}}{R_T}$. Parameters for (A–E) as in Fig 1.

bistable organization, a single disjoint interval was observed for all pulse train realizations (Fig 3C, green), since the receptor is irreversibly activated upon the first pulse. In the reversible bistable and monostable organization on the other hand, narrow skewed distributions were identified (realizations between 7–8 and 12 were identified, Fig 3C, yellow and blue). This reflects small degree of partitioning that results from immediate signal re-occurrences within the relaxation time of the system.

To increase the degree of integration of a monostable system in general, its kinetic parameters can be tuned to match the relaxation time to the transient memory length (Fig EV1, Materials and Methods). However in this case, a slow decay rate will greatly decrease the number of disjoint intervals in comparison with the one for critical organization, due to the absence of rapid reversibility in the response. Increasing the decay rate on the other hand will diminish the history-dependent signal integration, and the total duration of receptor activity will resemble the total duration of growth factor pulses. Thus, to avoid the trade-off between history-dependent signal integration and adaptation to the temporal partitioning of the signal, the response must exhibit transiently maintained steady levels of high receptor activity, followed by rapid reversibility to basal levels. Hence, simple relaxation kinetics of a monostable system cannot account for processing time-varying signals by receptor networks.

At the critical organization, the emergence of the dynamic memory is additionally complemented with a maximal increase in receptor activity to minimal growth factor dose that activates the system. Scanning the $P_{DNF,T}/R_T$ ratio demonstrated that the dynamic range of the response amplitude rapidly increases when transiting from the monostable towards the bistable regime, with a clear peak at the *SN* bifurcation (right to left, Fig 3E). These results therefore demonstrate that critical organization is crucial for robust responsiveness to and processing of time-varying growth factor signals.

### Molecular realization of transient memory in receptor activity

To understand how transient memory can be realized on a molecular level, we studied how transient receptor activity can be generated and maintained in the absence of stimulus using single-molecule reaction–diffusion simulation framework on a two-dimensional surface (Materials and Methods). By analogy to the main model (Equation 1), single receptor molecules (R) are susceptible to activation, and once active they can propagate their state via direct contact to other inactive molecules. The receptor molecules can become susceptible again following deactivation by the active $P_{DNF}$ molecules, with which they interact in double-negative feedback manner. To characterize the activity dynamics of the system, we use the basic reproduction number $R_0$ (Dietz, 1993) that reflects the transmission potential—the average number of newly activated molecules by a single active receptor molecule in the course of its active lifetime, i.e. before its deactivation. If $R_0 < 1$, the overall receptor activity in the system decays to a basal level (Fig 4A, top), whereas for $R_0 > 1$, the system exhibits supercritical behaviour and the receptor activity propagates in a branching fashion across the cell surface (Fig 4A, bottom).

To relate the $R_0$ values to the qualitative changes observed when crossing the saddle-node bifurcation points, we derived analytically the dependence of $R_0$ to the main parameter that determines the positioning of the system in the vicinity of the *SN* bifurcation point. For simplicity, we use $\gamma_{DNF}$ as a bifurcation parameter, which represents the specific reactivity of $P_{DNF}$ to $R_a$ ($\gamma_{DNF} \propto P_{DNF,a}/R_T$, Fig 1B). The activation transmission potential in every point of the macroscopic phase space can be therefore described as $R_0 \equiv \frac{\alpha_2 R_T(1-R_a)}{\gamma_{DNF} P_{DNF,T} P_{DNF,a}}$, where $\alpha_2 R_T(1-R_a)$ and $1/(\gamma_{DNF} P_{DNF,T} P_{DNF,a})$ are the average molecular activation rate and lifetime, respectively (kinetic parameters

given in Materials and Methods). The estimations show that when initiated at the basal state, hence with a fully susceptible population ($R_a = 0, P_{DNF,a} = \frac{k_1}{k_1+k_2}$), the critical threshold $R_0 = 1$ is crossed at the $\gamma_{DNF}$ value corresponding to $SN_1$ (Fig 4B bottom and middle) due to $R_0 = \frac{\alpha_2 R_T(k_1+k_2)}{\gamma_{DNF} P_{DNF,T} k_1} \propto \frac{1}{\gamma_{DNF}}$. For $\gamma_{DNF}$ smaller than $SN_1$, activity propagation is ensued as $R_0 > 1$. However, once the system reaches the high activity steady state, $R_0$ is maintained at 1 (Fig 4B top and middle), because mass conservation limits the number of susceptible molecules. The system loses the transmission potential for $\gamma_{DNF}$ values higher than $SN_2$.

Looking into the spatial organization, the reaction–diffusion simulations demonstrated that the high activity state was maintained over the whole integration time for bistable organization (Fig 4C, top and Movie EV2). For critical organization, the simulations show that local pockets of active receptors are sustained, that transiently maintain and thereby further propagate the high activity state across the surface to other local pockets (Fig 4C, middle and Movie EV3). This results in interchanging periods of inactivation and re-activation bursts around the "ghost" attractor state, manifested as prolonged receptor activity before the system settles to basal state. Thus, the proximity of $R_0$ to 1 together with the diffusion-induced local spatial inhomogeneities in $P_{DNF,T}/R_T$ concentration effectively increases the local transmission potential above the critical value 1 in certain areas, thereby generating local pockets of active receptor that transiently sustain and further propagate this state across the surface (Movie EV4). In contrast, for organization in the monostable region where $R_0 < 1$, the system rapidly converged to basal receptor activity state (Fig 4C, bottom and Movie EV5).

### Self-organized positioning at criticality by fluctuation sensing

Positioning at criticality uniquely endows receptor networks with the features necessary for processing time-varying growth factor signals. Recently, such organization has been experimentally revealed for the proto-oncogenic epidermal growth factor receptor (EGFR; Stanoev *et al*, 2018). Since it is unclear how the critical positioning is maintained over time despite other regulation mechanisms that reduce the EGFR concentration on the membrane, we use this example to investigate which molecular features of receptor tyrosine kinases are crucial for this. We note that we seek a qualitative rather than quantitative description of a possible dynamical mechanism that will maintain critical organization of receptor networks.

The EGFR activity dynamics is regulated by a four-component spatially distributed network, where the interactions of EGFR (R) with three specific membrane-associated enzymes—protein tyrosine phosphatases via a double negative ($P_{DNF}$, PTPRG), a negative feedback ($P_{NF}$, PTPN2) and a negative regulation ($P_{NR}$, PTPRJ), are coupled via the vesicular trafficking of the receptor (Fig 5A). Ligand-bound EGF receptors ($LR$) promote autocatalytic activation of ligandless receptors (red arrows; Reynolds *et al*, 2003; Tischer & Bastiaens, 2003; Baumdick *et al*, 2015, 2018) and thereby transfer information about the extracellular environment before they are internalized and degraded ($LR^E$; Schlessinger, 2002; Baumdick *et al*, 2015). The vesicular recycling, on the other hand, brings the internalized and deactivated ligandless EGFR ($R^E$) back to the plasma membrane, establishing the EGF signal processing network.

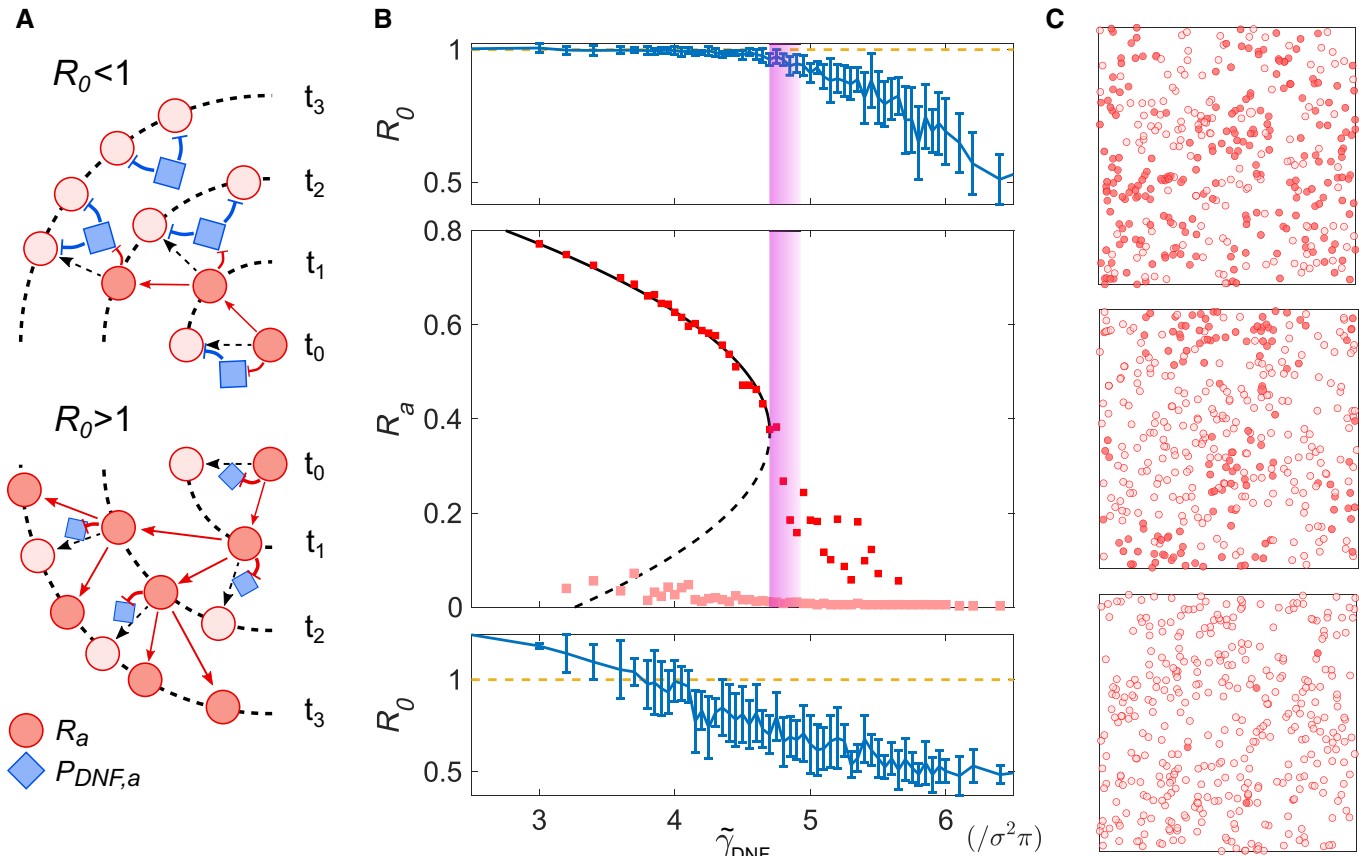

**Figure 4. Molecular realization of transient memory.**

A  Schematic representation of activity propagation in relation to microscopic single-molecule activation/inactivation dynamics. Top: diminished activity for $R_O < 1$. Bottom: propagated activity for $R_O > 1$.

B  Basic reproduction number for varying $\hat{\gamma}_{DNF}$ values estimated by single-molecule reaction–diffusion simulations. Middle: estimated high (dark red) and basal (light red) receptor activity states, black lines—estimated bifurcation profile; magenta—critical behaviour. Top/bottom plots: $R_O$ values around the higher (top) and basal (bottom) stable steady state. From $N = 10$ realizations for each $\hat{\gamma}_{DNF}$ value, mean $\pm$ SD is shown.

C  Snapshots of reaction–diffusion simulations for organization in bistable regime ($\hat{\gamma}_{DNF} = 4.55/(\sigma^2\pi)$, top), criticality ($\hat{\gamma}_{DNF} = 4.95/(\sigma^2\pi)$, middle) and monostable regime ($\hat{\gamma}_{DNF} = 6/(\sigma^2\pi)$, bottom), where $\sigma$ is the bimolecular interaction radius. Empty/filled circles denote inactive/active receptor molecules. $P_{DNF}$ molecules are not displayed for clarity. Details of the simulations and other parameters in Materials and Methods.

Numerical simulations using a two-compartment model that takes the trafficking-induced redistribution of receptors explicitly into account (Equation (2) in Materials and Methods) showed that the unidirectional internalization of the ligand-bound receptors decreases the EGFR concentration on the plasma membrane, which rapidly shifts the operation of the system into the monostable regime in the absence of compensating mechanisms. As a result, the dynamic range of the activation amplitude is decreased and there is loss of the transient memory (Fig 5B, compare blue to magenta lines). This implies that dynamical maintenance of EGFR concentration (the determining bifurcation parameter) at the plasma membrane is required for existence of transient memory. Thus, to counteract the shift of the system away from the *SN* bifurcation, the loss of EGFR receptors must be compensated with an increase of the recycling rate of ligandless receptors. A two-parameter bifurcation analysis shows how the position of the SN bifurcation depends on these two parameters (Fig 5C) and thus how $k_{rec}$ up-regulation, as a result of the decrease in total receptor concentration $R_T$, would

effectively retain the system in the vicinity of the *SN* for several subsequent growth factor pulses (Fig 5D, compare red with blue dots in Fig 5C). It should be noted, however, that the SN positioning asymptotically approaches a minimal receptor concentration below which the receptor recycling rate can no longer sufficiently compensate for the loss of receptors from the membrane (Fig 5C, dashed line). Additionally, the receptor recycling machinery may also impose an upper bound on $k_{rec}$, further limiting the resetting capacity. How many pulses are required to bring the system to the limit depends on the pulse duration and growth factor dose. Thereafter, the system will rely on synthesis of new receptors to re-establish critical positioning (not included in our model).

Dynamical regulation of the recycling rate and thereby maintenance of the critical organization require a mechanism for sensing receptor abundance to estimate the divergence from the saddle-node bifurcation point, coupled to an actuating mechanism to translate this positioning into a corresponding recycling rate. One possibility to realize the sensing of receptor abundance is through the

fluctuations in the active state of the receptor: the dominant frequency in basal EGFR activity fluctuations estimated as an average of multiple stochastic activity profiles (Materials and Methods) is minimal in the vicinity of the saddle-node bifurcation (Rogister *et al*, 2003; Fig 5E). This is because the temporal signature of the fluctuations depends on the alignment of the nullclines (Durstewitz, 2003), and thus in this case, by the concentration of the receptors that determines the positioning in parameter space. Therefore, an $k_{rec}$ actuating component that would readout these fluctuations through a low-pass filter can provide such coupling and maintain the critical organization of the system (Fig 5F). Previous experimental work has demonstrated that Akt, a serine–threonine kinase downstream of EGFR, promotes recycling of EGFR (Er *et al*, 2013; Stallaert *et al*, 2018), and furthermore, the Akt pathway has been implicated to serve as a low-pass filter (Fujita *et al*, 2010), suggesting it as a possible actuating component. Although the proposed implementation details of this dynamical mechanism remain to be experimentally probed, it represents a minimal sensing-actuation mechanism that dynamically maintains the system at the critical position based on the molecular details of the system.

## Discussion

Cells continuously sense and interpret time-varying chemical signals that reflect their changing environment, as a means to determine their functional output (Foxman *et al*, 1999; Welf *et al*, 2012; Skoge *et al*, 2014; Ryu *et al*, 2015). As receptors are the first layer of the cell that interprets the complex signals, their activity must exhibit transient memory features in order to enable sensing and interpreting of the temporal dependencies inherent in the complex signals. Here, we demonstrated that positioning of the receptor system in the vicinity of a saddle-node bifurcation provides a unique mechanism for processing time-varying growth factor signals. The metastable state that is thereby generated facilitates a dynamic transient memory which is a prerequisite for the computations. In contrast, for organization in the bistable regime—a classical example how "memory" is realized, receptor activity is sensitive neither to the number of growth factor pulses, nor to the temporal distribution of the signal.

It has been experimentally demonstrated that the EGFR sensing system is organized in a vicinity of a saddle-node bifurcation

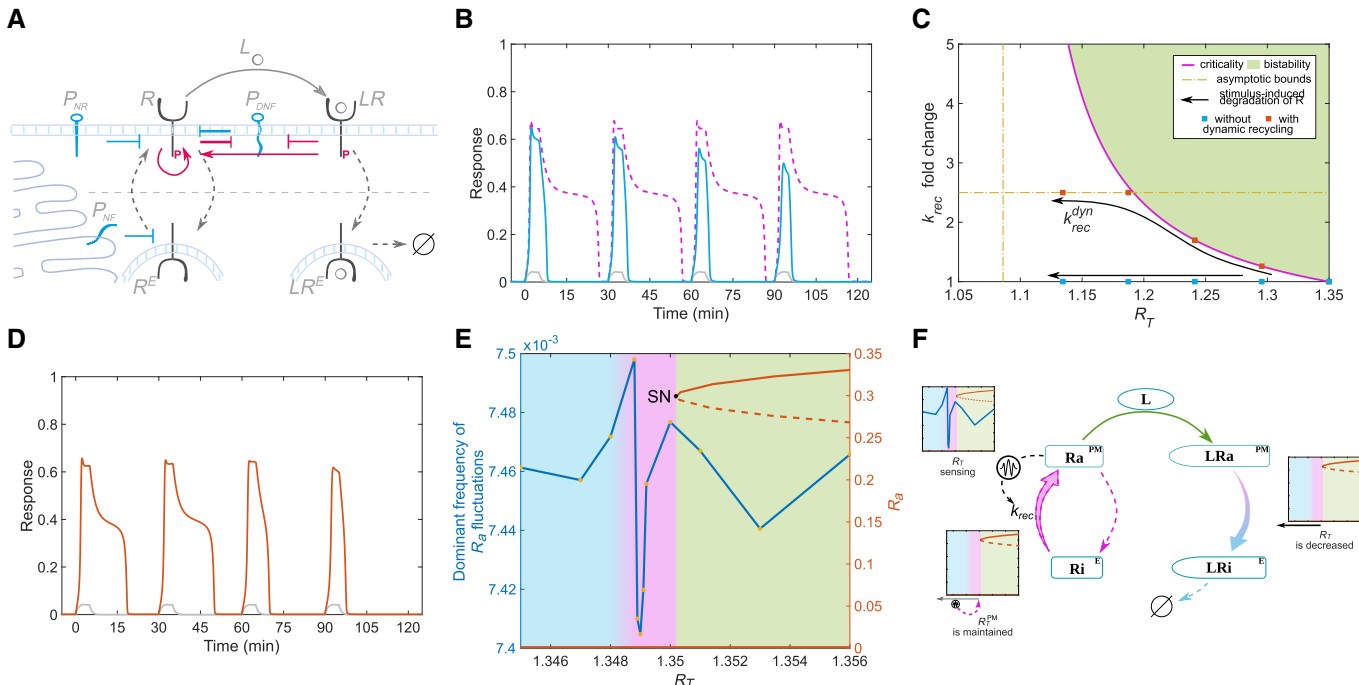

**Figure 5. Dynamical mechanism for self-organization at criticality.**

A   Schematic representation of the spatially distributed EGFR network. At the plasma membrane, ligandless EGFR (R) is coupled to PTPRG ($P_{DNF}$) in a double-negative feedback manner and negatively regulated by PTPRJ ($P_{NR}$). Activated receptors are then endocytosed in the perinuclear area ($R^E$), deactivated by PTPN2 ($P_{NF}$) and recycled back to the plasma membrane, which amounts to a spatially established negative feedback. Ligand (L) binding converts R to ligand-bound species (LR) that are internalized ($LR^E$) and subsequently degraded. *LR* promotes autocatalytic activation of R (red arrows).

B   Response of active (blue) and ligand-bound (grey) receptors at the cell surface, when degradation of ligand-bound receptors is explicitly considered for critical organization ($R_T$ = 1.3499; Materials and Methods, compared to the case without degradation (dashed magenta line).

C   Two-parameter ($k_{rec}$, $R_T$) bifurcation diagram depicting the positioning of the saddle-node bifurcation point $SN_2$ (magenta line). Green shaded area: bistability region and red dashed line: asymptotic limit of the saddle-node positioning.

D   Dynamically maintained transient memory in receptor activity (red) upon train of stimulus pulses. Grey line—fraction of ligand-bound receptors.

E   Continuation plot of receptor activity (red) as a function of total receptor concentration and dominant frequency of fluctuations in basal receptor activity (blue) estimated for specific $R_T$ values. Solid/dashed red line—stable/unstable steady states. Shading equivalent to Fig 1B.

F   Schematic representation of a fluctuation-sensing and actuating system that dynamically poises receptor concentration at the plasma membrane in the vicinity of the saddle-node bifurcation point. Parameters: Materials and Methods.

(Stanoev *et al*, 2018). However, given that the critical organization uniquely enables responsiveness to time-varying growth factor signals, we propose that this is likely a generic feature of receptor sensing networks. In particular, several members of the receptor tyrosine kinase (RTK) family, such as FGFR, PDGFR and VGFR, share similar response and trafficking dynamics with EGFR (Miaczynska, 2013) that helps us substantiate this hypothesis. As for EGFR, it has been also experimentally verified for the aforementioned RTKs that ligand-induced phosphorylation initiates production of reactive oxygen species (ROS) via PI3K-induced activation of the NADPH oxidase complex (Chen *et al*, 2007; Paletta-Silva *et al*, 2013; Ornitz & Itoh, 2015) at the plasma membrane, thus facilitating oxidation and subsequent reversible inactivation of the protein tyrosine phosphatases in the vicinity. This establishes the conditions for these RTKs to be embedded in a double-negative feedback network motifs, equivalently to the ROS-mediated EGFR-PTPRG toggle switch (Stanoev *et al*, 2018), ensuring the presence of a *SN* bifurcation. Even more, ligand-induced Akt activation which has been demonstrated to promote vesicular recycling of these receptors (Miaczynska, 2013; Stallaert *et al*, 2018) could, in a similar manner, maintain the sensing systems at the critical positioning. As these receptors play a key role during embryogenesis, adult physiology and pathophysiology (Miaczynska, 2013) during which processing time-varying growth factor signals is prevalent, it remains to be experimentally verified that similar to EGFR, these systems also have critical organization.

On the other hand, the largest family of signalling receptors, the G-protein coupled receptors (GPCRs), detect complex chemokine signals and transduce them to the actin network to direct cell migration (Chung *et al*, 2001). Within this complex environment, such as the movement of the social amoeba *D. discoideum* through travelling waves of chemoattractant, the dynamics of the receptor networks allows for the cell to maintain memory of the signals in order to sustain its directed forward motion (Skoge *et al*, 2014). In the case of GPCRs, a double-negative feedback and thereby associated *SN*s have been identified within the receptor network, between the direct downstream components that GPCRs activate, PI3K and PTEN (Matsuoka & Ueda, 2018). Similarly as in the EGFR sensing systems, the activity dynamics of these receptors is also tightly regulated by the endosomal trafficking. Even more, this regulation is mediated via PI3K, an Akt activator (Uchida *et al*, 2017). Thus, given that conditions for critical organization and maintenance in the vicinity of a *SN* bifurcation are also possible, it is likely that similar mechanism can confer responsiveness of GPCRs to time-varying growth factor signals.

The proposed conceptual framework therefore implies that the notion of "computation with stable attractors" (Turing, 1937; Hopfield, 1984; Hirsch & Baird, 1995) likely should be adapted for cellular processing of non-stationary signals. Crucial in the proposed information processing with state-dependent trajectories is the dynamic transient memory emerging at criticality. This ensures history-dependent signal integration, feature that was previously only attributed to large-scale neural networks (Beggs & Plenz, 2003; Haldeman & Beggs, 2005; Chialvo, 2006; Kinouchi & Copelli, 2006; Legenstein & Maass, 2007; Levina *et al*, 2007). We propose that this dynamical mechanism uniquely balances stability with overall cellular responsiveness, as pervasive for systems that operate in a continuously changing environment.

## Materials and Methods

### Modelling the R-P$_{DNF}$ toggle switch

With Equation (1), we model a minimal network motif that exhibits bistability, a double-negative feedback, using law of mass action. Both the receptor and the deactivating enzyme have active (R$_a$, $P_{DNF,a}$) and inactive (R$_i$, $P_{DNF,i}$) states, and their state transition rates are described by the model equations. Therefore, mass is conserved in the system and the total protein concentrations of both species ($R_T$, $P_{DNF,T}$) are constant parameters. This allows $R_i = 1 - R_a$ and $P_{DNF,i} = 1 - P_{DNF,a}$ to be expressed as fractions from the total protein concentrations. Since the fraction of ligand-bound receptors ($LR_a$) is mapped to the cell from the ligand concentration in the environment (Stanoev *et al*, 2018), it is considered as an input parameter in the system.

Autonomous, autocatalytic and ligand-bound-induced activation of ligandless $R_i$ ensue from bimolecular interactions with distinct rate constants $\alpha_{1-3}$, respectively. Other parameters are as follows: $k_{2/1}$—inactivation/activation constant ratio of $P_{DNF}$, $k_R$, $k_1$—kinetic constants that do not influence the steady-state values of the system, $\hat{\beta}_{DNF} = \beta_{DNF}R_T/k_1$ —receptor-induced regulation rate constant of $P_{DNF}$ and $\hat{\gamma}_{DNF} = \gamma_{DNF}P_{DNF,T}/R_T$—specific reactivity of the enzyme towards the receptor, thus proportional to the local effective $P_{DNF,T}/R_T$ ratio. In the analysis, we refer to changes of $P_{DNF,T}/R_T$ when numerically $\hat{\gamma}_{DNF}$ is varied.

For simulations with growth factor pulses in Figs 1E and 2A–C, and 3A–D, binding/unbinding of ligand $L_T$ to modulate $LR_a$ was introduced. Thus, $-k_{on}R_aL_T + \frac{1}{2}k_{off}LR_a$ was added as additional term to the differential equation of $R_a$, and the dynamics of $LR_a$ was modelled with $\frac{dLR_a}{dt} = k_{on}(R_a + R_i)L_T - k_{off}LR_a$. For the simulations in the Movie EV1, a stochastic differential equation model was constructed from Equation (1) by adding a multiplicative noise term $\sigma X_i(1 - X_i)dW_t$, where $\sigma = 0.05$, $dW_t$ is the Brownian motion term and $X_i(1 - X_i)$ is the state-dependent function for each variable $i$ that accounts for mass conservation and normalization of the variables. The model was solved with $\Delta t = 0.01$ using the Euler solver from the "Financial Toolbox" in MATLAB. The parameters corresponding to Figs 1–3 are as follows: $\alpha_1 = 0.0017$, $\alpha_2 = 0.3$, $\alpha_3 = 1.0$, $\hat{\beta}_{DNF} = 36.0558$, $\hat{\gamma}_{DNF} \in \{2.5, 2.957, 3.5, 4.3\}$ (bistability, criticality, reversible bistability, monostability), $k_1 = 0.01$, $k_{2/1} = 0.5$, $K_R = 0.8$, $k_{on} = 0.003$, $k_{off} = 0.01668$. The L$_T$ amplitude during the pulse was set to produce $LR_a = 0.15$ in steady-state. In all figures, Response = $R_a + LR_a$.

### Quasi-potential landscape computation

The numerical computation of the quasi-potential landscapes, corresponding to the phase space diagrams in Fig 2A–C, was conducted using an approach adapted from the one in Bhattacharya *et al* (2011). Multiple trajectories were calculated starting from initial conditions distributed on a grid in the phase space. Rate of change in the quasi-potential (initiated arbitrarily to 0) was calculated by $-\frac{dR_a}{dt}\frac{dR_a}{dt} - \frac{dP_{DNF,a}}{dt}\frac{dP_{DNF,a}}{dt}$ in every iteration, and the quasi-potential was integrated by the ODE solver together with the system trajectory integration. Quasi-potentials

of trajectories converging to the same attractor were aligned to match at the steady-state level. Quasi-potentials in basins of different attractors were subsequently aligned at the initial points, close to one another, of trajectories that converge to the different respective attractors, i.e. at the separatrix points. Additionally, neighbouring separatrix pairs were weighted by the angle between their derivatives ($\theta$), according to the formula: $\frac{1}{2}(1 - \cos\theta)$. This gives greater weight to diverging pairs, effectively aligning the separatrix quasi-potential values at the saddle steady-state point. The quasi-potential landscape at every point in phase space was finally estimated by interpolation from the aligned quasi-potential values of all of the trajectory points.

### Estimation of the basic reproduction number using single-molecule reaction–diffusion framework

We considered a two-dimensional domain representing the plasma membrane containing reacting and diffusing single molecules. The spatial coordinates of the molecules were updated using Brownian dynamics, and time was discretized to intervals of length $\Delta t$. First-order unimolecular reactions occur spontaneously with probability $\tilde{k}\Delta t$, where $\tilde{k}$ is the intrinsic reaction rate constant. Second-order bimolecular reactions on the other hand are modelled using the Doi method (Doi, 1976), following the Smoluchowski single-particle framework for describing diffusion influenced reactions (Smoluchowski, 1917). An interaction takes place between two molecules that have diffused within a proximity distance $\sigma$ of each other, and a reaction ensues with a probability $\tilde{g}\Delta t$, where $\tilde{g}$ is the microscopic bimolecular reaction rate constant. $\sigma$ is of order of the molecule radius. In the rare event that a substrate molecule is in proximity of $n > 1$ other enzyme molecules, the reaction takes place with probability $1 - (1 - \tilde{g}\Delta t)^n$, assuming any of the enzyme molecules can affect the state of the given substrate molecule. Formation of the product proceeds immediately upon successful bimolecular enzyme–substrate interaction, i.e. the state of the substrate molecule is directly changed. To model the interactions between $R$ and $P_{DNF}$ (Fig 1A), we assumed that both particles diffuse across the 2D domain with equal diffusion rates $D_R = D_{P_{DNF}} = D = 0.1\mu m^2/s$. The interaction radius $\sigma$ was set to $2\rho$, where $\rho = 10nm$ is the molecule radius. Receptor and $P_{DNF}$ molecules, with density of 60 and 80 molecules per square micrometre, respectively, were randomly deployed on a $3.5\mu m \times 3.5\mu m$ or $2.5\mu m \times 2.5\mu m$ surface area and allowed to diffuse using Brownian dynamics with periodic boundary conditions. $\Delta t$ was set to $1 \times 10^{-4}$ s to ensure that $\sqrt{4(2D)\Delta t} \leq \sigma$, i.e. any interaction between two molecules that come in proximity is detected, and also to ensure that no reaction probability is greater than one. The state transitions of $R$ and $P_{DNF}$ occur in accordance with the macroscopic description—Equation (1), only in the absence of external input ($LR_a = 0$). The microscopic rate constants are generally proportional to the ones in our main ODE model—$\tilde{\alpha}_1 = 0.0017/(\sigma^2\pi)$, $\tilde{\alpha}_2 = 0.3/(\sigma^2\pi)$, $\tilde{\beta}_{DNF} = 9/(\sigma^2\pi)$, $\tilde{k}_1 = 5$, $\tilde{k}_2 = 0.5$. They were set to produce faster kinetics, due to numerical and data storage constraints. By analogy to the macroscopic bifurcation analysis ($\tilde{\gamma}_{DNF} \propto P_{DNF,T}/R_T$), $\tilde{\gamma}_{DNF}$ was varied between 0 and $9/(\sigma^2\pi)$ ($P_{DNF,T}$ and $R_T$ were kept constant) to modulate the specific reactivity of $P_{DNF,a}$ towards $R_a$. To calculate the basic reproduction number $R_0$, the number of

substrate receptor molecules $R_{0,j}(t)$ that each $R_{a,j}$ molecule successfully activated within its activation lifetime was recorded, after that molecule has been previously activated itself at time $t$. $R_0$ was calculated as an average of these figures within a certain time interval (approximately 200ms). Theoretically, $R_0$ depends on the probability of activating a susceptible receptor molecule and the duration of the activity lifetime. Neglecting the effect from autonomous activation of R (with low rate $\alpha_1$), we arrive at the following definition $R_0 \equiv \frac{\alpha_2 R_T(1-R_a)}{\gamma_{DNF}P_{DNF,T}P_{DNF,a}}$. From Equation (1), it is straightforward to determine the basal activity stable steady state as $R_a = 0, P_{DNF,a} = \frac{k_1}{k_1+k_2}$ and thus $R_0 = \frac{\alpha_2 R_T(k_1+k_2)}{\gamma_{DNF}P_{DNF,T}k_1}$. By employing linear stability analysis, we could determine that the condition for stability of this steady state is indeed $R_0 < 1$. On the other hand, it follows readily from the first equation that $R_0 = 1$ around the high activity stable steady state, where $R_a \neq 0$. There we could also find that $\gamma_{DNF} = \frac{\alpha_2 R_T^2}{k_1 P_{DNF,T}}(1 - R_a)\left(\frac{k_1+k_2}{R_T\beta_{DNF}} + R_a\right)$, and thus, there is approximately a quadratic dependence between $\tilde{\gamma}_{DNF}$ and $R_a$. Quadratic form was therefore used in Fig 4B, middle, to estimate the bifurcation profile from the extensively occupied high and basal receptor activity states. To estimate the extensively occupied high and basal receptor activity states from the trajectories in $R_a$-$P_{DNF,a}$ phase space, Gaussian mixture distribution was fitted to the data with two components and data points were pruned iteratively with $90^{th}$ percentile cut-off until convergence.

### Compartmental model of spatial–temporal EGFR activation regulation

The experimentally derived EGFR-PTP network (Stanoev *et al*, 2018) was implemented using a two-compartment model that includes explicitly the vesicular trafficking between the plasma membrane and the endosomal compartments (Fig 5A), as described using the following system of ODEs:

$$\begin{aligned}
\frac{dR_a^{PM}}{dt} &= R_T R_i^{PM}\left(\alpha_1 R_i^{PM} + \alpha_2 R_a^{PM} + \alpha_3 LR_a^{PM}\right) - \gamma_{DNF}P_{DNF,T}P_{DNF,a}R_a^{PM} \\
&\quad - \gamma_{NR}P_{NR,T}R_a^{PM} - k_{in}R_a^{PM} - k_{on}R_a^{PM}L_T + 0.5k_{off}LR_a^{PM} \\
\frac{dR_i^{PM}}{dt} &= -R_T R_i^{PM}\left(\alpha_1 R_i^{PM} + \alpha_2 R_a^{PM} + \alpha_3 LR_a^{PM}\right) + \gamma_{DNF}P_{DNF,T}P_{DNF,a}R_a^{PM} \\
&\quad + \gamma_{NR}P_{NR,T}R_a^{PM} - k_{in}^i k_{in}R_i^{PM} + k_{rec}R_i^E - k_{on}R_i^{PM}L_T \\
&\quad + 0.5k_{off}LR_a^{PM} \\
\frac{dP_{DNF,a}}{dt} &= k_1\left[(1 - P_{DNF,a}) - k_{2/1}P_{DNF,a} - \beta_{DNF}P_{DNF,a}R_T\left(R_a^{PM} + LR_a^{PM}\right)\right] \\
\frac{dLR_a^{PM}}{dt} &= k_{on}\left(R_a^{PM} + R_i^{PM}\right)L_T - k_{off}LR_a^{PM} - k_{deg}k_{in}LR_a^{PM} \\
\frac{dR_a^E}{dt} &= k_{in}R_a^{PM} - \gamma_{NF}P_{NF,T}R_a^E \\
\frac{dR_i^E}{dt} &= k_{in}^i k_{in}R_i^{PM} + \gamma_{NF}P_{NF,T}R_a^E - k_{rec}R_i^E \\
\frac{dLR_i^E}{dt} &= k_{deg}k_{in}LR_a^{PM}
\end{aligned}$$

$$(2)$$

$P_{DNF}$ (PTPRG), $P_{NF}$ (PTPN2) and $P_{NR}$ (PTPRJ) represent the major protein tyrosine phosphatases that regulate EGFR ($R$, $LR$) activity, $\gamma_x$—specific reactivity that each $P_X \in \{P_{DNF}$ (PTPRG), $P_{NF}$ (PTPN2),

$P_{NR}$ (PTPRJ)} has towards EGFR. $k_{in}$, $k_{rec}$ and $k_{deg}$ denote receptor internalization, recycling and degradation rate constants, respectively, i,a—inactive and active species, and E, PM—endosomal and plasma membrane species. $k_{deg} = 0$ for the magenta profile in Fig 5B. $LR_i^E$ is the accumulated endosomal ligand-bound EGFR and thus the effective degraded fraction of EGFR. For the self-organizing criticality model (Fig 5D), the dynamically maintained $k_{rec}^{dyn} = \frac{R_T - R_{T,asymp}}{R_T(1 - LR_i^E) - R_{T,asymp}} k_{rec}$, where $R_{T,asymp} = 1.086$ is the lower bound asymptotic value of $R_T$ in dependence to $k_{rec}$ (dashed line, Fig 5C). Saturation level of 2.5 for the multiplier term is also assumed, beyond which the recycling rate can no longer increase. Parameters are as follows: $\gamma_{DNF} = 3.0$, $\gamma_{NF} = 3.0$, $\gamma_{NR} = 0.001$, $\beta_{DNF} = 36.0558$, $k_{in} = 0.02$, $k_{rec} = 0.042$, $k_{in}^i = 0.2$, $k_{deg} = 0.2$, $R_T = 1.3499$, $L_T = 0.2926$ and $P_{DNF,T} = P_{NF,T} = P_{NR,T} = 1.0$. Other parameters are the same as in Fig 1 (Materials and Methods).

## Model calibration

The parameters in the model (Figs 1A and 5) were described in Stanoev *et al* (2018) and calibrated with the single-cell dose–response data described therein, from where the topology of the sensing network (Fig 5A) was derived. We convert from dimensionless time to minutes by equating the EGFR phosphorylation kinetics and duration in the simulations using the kinetic parameters to the experimental values in Stanoev *et al* (2018). The parameters for the microscopic dynamics in the single-molecule reaction–diffusion simulations were set to scale the macroscopic ODE parameters and set to produce faster kinetics due to numerical reasons, as described in the corresponding section.

## Stochastic simulations

To model the contribution of random fluctuations in the activity dynamics of the network constituents, a multiplicative noise term was added to the first equation in the ODE system (Equation 2): $g(R_a^{PM})\xi(t)$. $\xi(t)$ is a Gaussian white noise with zero mean and temporal correlation $<\xi(t)\xi(t')> = \sigma_a^2\delta(t - t')$, $\delta(t - t')$ is the Dirac delta function and $\sigma_a^2$ is a constant that characterizes the noise intensity The multiplicative noise term is interpreted according to Stratonovich (Gardiner, 2009), as a stochastic interpretation for a realistic noise with small temporal autocorrelation (García-Ojalvo & Sancho, 2012). This noise term can incorporate both intrinsic and extrinsic sources (Koseska *et al*, 2007). We establish the function $g(R_a^{PM})$ by means of simple approximation, assuming that the relative fluctuations scale is the inverse square-root of the amount of active protein. Such scaling is generic for many stochastic processes (e.g. Poisson processes or birth–death processes) and provides means to investigate the implications of fluctuations on the dynamics of biochemical networks in general (Koseska *et al*, 2007). To avoid negative values of the protein concentrations due to the introduced stochasticity in the system, a custom-made adaptive step size algorithm (Kloeden & Platen, 1992) employed to Euler integration scheme was developed in C. $N = 50$ realizations of stochastic time series were simulated with $\sigma_a^2 = 8 \times 10^{-4}$ and for each of them the dominant frequency in basal receptor activity was extracted by computing the dominant mode of the wavelet power spectrum using the "WaveletComp" package in R (Roesch & Schmidbauer, 2018). The average dominant frequency from these realizations is plotted in Fig 5E.

### Exponential decay model

To model a receptor activity profile that follows an exponentially decaying relaxation process, we implemented:

$$\frac{dR_a}{dt} = I\alpha(R_T - R_a) - \beta R_a \tag{3}$$

where $I$ denotes the growth factor pulse, $\alpha$ and $\beta$—the activation and decay rate constants, respectively, and $R_T$—the total protein concentration. For Fig EV1, five distinct $\beta$ values are used, for which the degree of history-dependent signal integration and the degree of partitioning in response to complex growth factor train pulses are estimated. For all model realizations, $R_T$ was set such that the temporal receptor activity profiles from the exponential decay models reached the same maximal steady state amplitude as the one for organization at criticality. Parameters: $\alpha = 0.023$; $(\beta, R_T) \in \{(1.5 \times 10^{-4}, 0.8206), (5.5 \times 10^{-4}, 0.8351), (0.001, 0.8513), (0.0035, 0.9416), (0.01, 1.1765)\}$.

The numerical bifurcation analysis was performed using the XPP/AUTO software (Ermentrout, 2016). All simulations except where explicitly noted were performed using custom-made code in MATLAB (MATLAB and Statistics Toolbox Release R2018a, The MathWorks, Inc., Natick, Massachusetts, United States).

## Data and software availability

All data and code used in this study are available in the following location: https://github.com/astanoev/ReceptorNetworks.

**Expanded View** for this article is available online.

### Acknowledgements
The authors thank Philippe Bastiaens for numerous discussions and suggestions that were crucial for the development of this work, as well as for critically reading the manuscript.

### Author contributions
AK conceptualized the study. AS performed most of the analytical derivations, numerical simulations and bifurcation analysis with help of APN and AK. AS and AK interpreted the results and wrote the manuscript with help of APN.

### Conflict of interest
The authors declare that they have no conflict of interest.

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
