## [Review Process File · Molecular Systems Biology]

Organization at criticality enables processing of time-varying signals by receptor networks

Angel Stanoev, Akhilesh P. Nandan, Aneta Koseska.

Review timeline:

Submission date:	20 th February 2019
Editorial Decision:	3 rd May 2019
Appeal received:	13 th May 2019
Editorial Decision:	10 th June 2019
Revision received:	20 th September 2019
Editorial Decision:	1 st December 2019
Revision received:	17 th December 2019
Accepted:	23 rd December 2019

Editor: Thomas Lemberger

Transaction Report:

1st Editorial Decision

3rd May 2019

Thank you again for submitting your work to Molecular Systems Biology and I greatly apologize again for the lengthy process. We have heard back from two of the three referees who accepted to review your study. Rather than delaying further the process, I prefer to make a decision now with the available reports. As you will see from the reports below, the referees find the general topic of the study of potential interest. They raise however substantial concerns on your work, which, I am afraid to say, preclude its publication in Molecular Systems Biology.

The reviewers do recognize that the study of signaling dynamics out of equilibrium is important. They also appreciate that you suggest that the proposed mechanisms might be general. They were however not convinced that the claims of the paper are well supported and sufficiently justified theoretically. While the proposed 'self-organization' mechanism is seen as plausible, the reviewers note that it is not supported by any data and remains therefore largely speculative. Finally, demonstrating the generality of the existence and the importance of a critical transition in biochemical pathways other than the specific example already reported in the previous Cell Systems paper would be crucial to provide a decisive advance.

Under these circumstances, I see no other choice than to return the manuscript with the message that we cannot offer to publish it. I am very sorry not to be able to bring better news on this occasion.

It is of course possible that with additional experimentation and analyses, the conclusions might be better supported experimentally and theoretically, in particular with regard to the generality and broad relevance of the proposed mechanisms. We would thus not be opposed to consider a new submission that would be significantly extended along these lines to address the major issues delineated above and by the reviewers. This would have a new number and receipt date. We recognise that this may involve further experimentation and analysis, and we can give no guarantee about its eventual acceptability. However, if you do decide to follow this course then it would be helpful to enclose with your re-submission an account of how the work has been altered in response

to the points raised in the present review. We would also be happy to discuss the matter further over the phone if you feel it would be helpful.

REFEREE REPORTS

Reviewer #1:

A large focus of dynamical systems theory is the identification of qualitative behavior of systems around steady states. Unfortunately, biological systems are often far from steady state. In this paper Stanoev et al analyze receptor-mediated signaling using EGFR as a model for benefits of non-equilibrium behavior. Overall the work is interesting, however, I had a hard time understanding the theoretical support for many of the claims made here. My recommendation is for the authors to clearly define all the different attributes they associate with the system and specifically demonstrate all their claims based on these definitions.

For example, I am having a hard time understanding why a system that has a critical point between monostable and bistable shows "plasticity and robustness". Figure 1 title is "Plasticity and robustness to non-stationary cues emerges at criticality." However, nothing in the figures specifically addresses neither plasticity or robustness. It just shows the "transient memory" that comes from the longer transient cells go through.

Similarly, the author claims about the special positioning of critical point that: "In this organization, the system does not process information using the stable attractors associated with specific states (0-1 case), but rather the information is maintained via the transient memory and interpreted through the phase-space trajectories". I really don't understand what kind information processing is happening there... this is just a long transient behavior.

In figure 3 the author claim that the system shows "Signal integration and optimal information processing". I don't understand what is the support for these claims. What is the optimality criteria used? What signals are being integrated? Why is this unique to systems that pass through a critical point?

Finally, the speculative model in figure 4 that suggests that it would be good for the system to sense fluctuation in concentration is 1) not supported by any data 2) not clear to me why it is better than direct sensing of concentration. Why is it less sensitive to noise?

Minor points:

1. Figure 1 is very hard to follow with all the arrows, potentials, etc. I recommend simplifying or revising it.
2. To emphasize the generalities of their findings and universality of the approach the authors analyzed the receptor system using an epidemic model and also include a model of a leaky neuron. In my opinion (and I understand that this could reflect personal preference) these are distracting and should be removed from the paper.
3. The authors use a lot of dynamical systems jargon. It will be more accessible to broader readership if they try to reduce the use of such jargon

Reviewer #2:

The manuscript by Stanoev et al investigates the properties of an input-output response near a dynamical transition between a monostable and a bistable state. The authors focus on a toggle-switch-type signaling network inspired by the EGFR network. Their main result is that, after switching from the low to high state, the transition back to the low state (i) is fast if the high state disappears, or (ii) may never occur if the high state is retained, but (iii) can be slow following a transient if the high state is "almost" retained (the so-called "ghost attractor) which occurs near the dynamical critical point. They show that this behavior persists in a spatially resolved version of their receptor signaling model, as well as in an unrelated neuronal model, and they make an argument for how the EGFR network might keep itself near this critical point using self-organization (i.e.

corrective feedback).

Understanding general principles of memory and plasticity in signaling networks is an important goal that is highly relevant to the field of molecular systems biology. However, there are concerns with this manuscript that make it unclear whether it reaches this goal. The concept of a ghost attractor is intuitive and well studied. Therefore, the novelty of this study would potentially be to explain the mechanism more deeply and/or demonstrate that it occurs across a wide range of biochemical systems. I am not sure that it sufficiently achieves either of these goals. Specifically:

1. It is expected that both sensitivity and noise are maximal near a critical transition. However, the authors find that sensitivity is maximal (Fig 3E) but noise is maximal on one side and not the other (Fig 3F). What is the mechanism for this? Currently it is unclear because the results are purely numerical.
2. Relatedly, in the EGFR model the authors claim that the noise is lowest near the transition (Fig 4E). However, the plot of noise vs the bifurcation parameter (blue curve in Fig 4E) is wildly fluctuating, and I am not convinced that it has a clear minimum. This underscores the confusion about the role of noise near the transition.
3. The authors aim to present a plausible mechanism for the EGFR system to retain itself near the ghost attractor (Fig 4). However, the mechanism does not appear to succeed after the second pulse (Fig 4D). This makes it hard to understand whether their self-organization argument applies.
4. The proximity of the EGFR system to its critical transition appears to have been demonstrated in a previous study (Cell Systems 2018). Therefore the purpose of the present work, beyond elaborating on the advantages of this proximity, seems to be to demonstrate its generality. However, apart from the EGFR system, the authors only appeal to a neuronal model, where transient memory and plasticity is relatively well established. Therefore, I am not convinced that the authors' have sufficiently demonstrated that their findings are as general within biochemical networks as they claim.

Reviewer #1:

A large focus of dynamical systems theory is the identification of qualitative behavior of systems around steady states. Unfortunately, biological systems are often far from steady state. In this paper Stanoev et al analyze receptor-mediated signaling using EGFR as a model for benefits of non-equilibrium behavior. Overall the work is interesting, however, I had a hard time understanding the theoretical support for many of the claims made here. My recommendation is for the authors to clearly define all the different attributes they associate with the system and specifically demonstrate all their claims based on these definitions.

For example, I am having a hard time understanding why a system that has a critical point between monostable and bistable shows "plasticity and robustness". Figure 1 title is "Plasticity and robustness to non-stationary cues emerges at criticality." However, nothing in the figures specifically addresses neither plasticity or robustness. It just shows the "transient memory" that comes from the longer transient cells go through.

In Figure 1 we have demonstrated the principle how cell surface receptors can maintain memory in receptor activity even after growth factor signal has been removed, which confers robust signaling response to signal changes.

Plasticity on the other hand refers to the capability of the receptor to sense upcoming growth factor signals. As demonstrated, this is uniquely enabled at the critical transition, since the memory in receptor activity comes about via a metastable state (presence of the "ghost" attractor).

It was misunderstood that this transient memory is not a consequence of the kinetic parameters of the system but results from the closeness to the saddle-node bifurcation, and thereby has completely different features than typical transient behavior. In this case, it represents a mechanism for memory in receptor activity that is dynamic and thereby enables sensing signals while in the memory regime.

In contrast, organization in the bistable regime will only provide robustness to the system such that the receptor activity will be continuously maintained after the first signal has been removed. Since the system finds itself in a stable attractor, it can not reset to the basal receptor phosphorylation levels and therefore it cannot be responsive to upcoming signals. For organization in the monostable regime the situation is reversed – the system has single steady state and therefore phosphorylation profile of the receptor will always closely follow the growth factor signal, meaning that the system will be responsive. However, it cannot retain the phosphorylation after the signal has been removed, and thereby it cannot display robustness in the signaling response to signal changes.

Similarly, the author claims about the special positioning of critical point that: "In this organization, the system does not process information using the stable attractors associated with specific states (0-1 case), but rather the information is maintained via the transient memory and interpreted through the phase-

space trajectories". I really don't understand what kind information processing is happening there... this is just a long transient behavior.

The information processing is not a simple output reflection of the input signal in steady state, but has an additional temporal component that provides robustness to signal changes. We strongly disagree with the referee that this is just a strong transient behavior. The observed transient memory is, as previously noted, not a long (exponential) decay to basal state but the system occupies an intermediate, metastable state.

In contrast, when the system is organized in the bistable regime, the system has two stable steady states, basal receptor phosphorylation level (0 level) and high receptor phosphorylation level (level 1). Thus, stimulating with growth factor will push the system from 0 state to 1 state. Upon removing the growth factor signal however, the receptor will remain in state 1, since this is a stable attractor, thus maintaining the information that a growth factor has been present through continuous prolonged receptor phosphorylation.

In figure 3 the author claim that the system shows "Signal integration and optimal information processing". I don't understand what is the support for these claims. What is the optimality criteria used? What signals are being integrated? Why is this unique to systems that pass through a critical point?

In Figure 3 we have demonstrated 3 crucial receptor response properties that together constitute optimal processing of information about extracellular growth factors. We have demonstrated i) switch-like receptor phosphorylation response to increased doses of growth factors, that enables cells to maintain threshold for activation and thereby filter spurious signals, while rapidly and robustly activating to the threshold growth factor dose (Fig. 3d), ii) the dynamic range of the receptor activation for the input dose that activates the system is maximal for criticality organization (Fig. 3e), and iii) the probability for spurious receptor activation using single molecule simulations is minimal also in this organization (Fig. 3f). All these properties come about due to the positioning of the system in the vicinity of the SN bifurcation.

In terms of the signal integration – we tested the capability of the system to sense streams of signals, while the system is occupying the metastable memory state. As demonstrated, the system can respond to the second signal by prolonging the activity of the receptor, and thereby it can effectively integrate the response to the first with the response to the second signal. In contrast, when in reversible bistable organization – the receptor responds independently to each signal in the stream, and in irreversible bistable organization – the receptor stops responding after the first signal as the system is stuck in the stable attractor of high receptor phosphorylation (Figure 3a).

Finally, the speculative model in figure 4 that suggests that it would be good for the system to sense fluctuation in concentration is 1) not supported by any

data 2) not clear to me why it is better than direct sensing of concentration. Why is it less sensitive to noise?

In Fig.4 we proposed a mechanism how the system can maintain itself exactly at this critical point between the mono- and bistable mode of operation. We are not aware of any example in the literature where the cell can sense and regulate its own receptor concentration on the plasma membrane. We therefore propose that this could be possible through a fluctuation sensing mechanism that activates receptor recycling via a low-pass filter to maintain positioning of the system in the vicinity of the saddle-node bifurcation. This mechanism is probable since this adaptive feedback has the highest sensitivity to be triggered in this region.

This represents a possible hypothesis for the EGFR system that we base on previous experimental findings: that the EGFR system sits on the transition between mono- and bistable mode of operation (Stanoev et al., Cell Systems, 2018), that Akt activity promotes EGFR recycling (Stallaert et al., Sci. Signaling 2018) and that Akt is embedded in a low-pass filter (Fujita et al., Sci. Signaling 2010).

Minor points:

1. Figure 1 is very hard to follow with all the arrows, potentials, etc. I recommend simplifying or revising it.
2. To emphasize the generalities of their findings and universality of the approach the authors analyzed the receptor system using an epidemic model and also include a model of a leaky neuron. In my opinion (and I understand that this could reflect personal preference) these are distracting and should be removed from the paper.
3. The authors use a lot of dynamical systems jargon. It will be more accessible to broader readership if they try to reduce the use of such jargon

Reviewer #2:

The manuscript by Stanoev et al investigates the properties of an input-output response near a dynamical transition between a monostable and a bistable state. The authors focus on a toggle-switch-type signaling network inspired by the EGFR network. Their main result is that, after switching from the low to high state, the transition back to the low state (i) is fast if the high state disappears, or (ii) may never occur if the high state is retained, but (iii) can be slow following a transient if the high state is "almost" retained (the so-called "ghost attractor) which occurs near the dynamical critical point. They show that this behavior persists in a spatially resolved version of their receptor signaling model, as well as in an unrelated neuronal model, and they make an argument for how the EGFR network might keep itself near this critical point using self-organization (i.e. corrective feedback).

Understanding general principles of memory and plasticity in signaling

networks is an important goal that is highly relevant to the field of molecular systems biology. However, there are concerns with this manuscript that make it unclear whether it reaches this goal. The concept of a ghost attractor is intuitive and well studied.

Although the concept of a ghost attractor is known (Strogatz and Westervelt, PRB, 1989), to the best of our knowledge, it has not been proposed as a possible mechanism that can generate dynamic memory and thereby confer balance between plasticity and robustness in networks in general. Currently, there is no dynamical mechanism that explains how robustness and plasticity – core properties of living systems, can be unified through the dynamics of the underlying biochemical networks. The current models of multistability and switching between stable attractors, as we demonstrate here, are limited in this regard. Moreover, the concept of a ghost attractor has been somewhat investigated in neuronal networks (Izhikevich, 2007), however, even there, the possibility to generate a dynamic memory that can serve to integrate multiple signals has not been proposed. Therefore, we strongly believe that the proposed dynamical mechanism for processing time-varying signals is novel and brings a basic principle how cells process information from non-stationary environments. This is especially important for example for cells that sense and move in complex environments – as we will elaborate in the discussion part in the amended version.

Therefore, the novelty of this study would potentially be to explain the mechanism more deeply and/or demonstrate that it occurs across a wide range of biochemical systems. I am not sure that it sufficiently achieves either of these goals.

The principles of processing time-varying signals with dynamic memory as proposed in our manuscript relies only on the closeness to a saddle-node bifurcation. We demonstrate that this mechanism provides optimal processing of non-stationary signals and will therefore apply to any biochemical system that displays bistability and is organized in the vicinity of the saddle-node bifurcation. Due to the large number of biochemical networks that display bistable behavior (e.g. various receptor networks, the MAPK network, gene regulatory networks during development etc.) the potential to expand this study with multiple other examples that are also governed by different type of dynamics (i.e. Hill-type dynamics) is very large. We will therefore incorporate / discuss other biochemical networks that process non-stationary signals and for which bistability has been proposed/demonstrated.

Specifically:

1. It is expected that both sensitivity and noise are maximal near a critical transition. However, the authors find that sensitivity is maximal (Fig 3E) but noise is maximal on one side and not the other (Fig 3F). What is the mechanism for this? Currently it is unclear because the results are purely

numerical.

In Figure 3F we show the propensity for spontaneous activation, and not the amplitude of the noise.

Even more, in contrast to critical phenomena characterizing phase transitions, the features of the system in the critical vicinity of different bifurcations such as saddle-node or pitchfork bifurcation are different. It has been experimentally demonstrated for example for an external-cavity laser diode (Rogister et al., PRE 2003) that the frequency of the fluctuations is minimal in the vicinity of the transition due to the saddle-node ghost, a principle that we also use and demonstrate in Fig.4.

2. Relatedly, in the EGFR model the authors claim that the noise is lowest near the transition (Fig 4E). However, the plot of noise vs the bifurcation parameter (blue curve in Fig 4E) is wildly fluctuating, and I am not convinced that it has a clear minimum. This underscores the confusion about the role of noise near the transition.

We demonstrate that the frequency of the fluctuations rather than the noise amplitude itself has a minimum in the vicinity of the transition due to the saddle-node ghost, as already noted in the previous answer. We therefore postulate that a low-pass filter mechanism enables fluctuation sensing and maintains positioning in the SN vicinity.

3. The authors aim to present a plausible mechanism for the EGFR system to retain itself near the ghost attractor (Fig 4). However, the mechanism does not appear to succeed after the second pulse (Fig 4D). This makes it hard to understand whether their self-organization argument applies.

Indeed, the mechanism can only provide a short-term maintenance at the SN bifurcation by tipping the plasma membrane vs endosomal receptor concentration balance to counteract the receptor removal from the membrane. This is dynamically limited, since the SN positioning asymptotically approaches a receptor concentration below which the recycling rate cannot compensate for the loss of receptors on the membrane. Additionally, the receptor recycling machinery might also impose an upper bound on the recycling rate, further limiting the resetting capacity. The cell will therefore rely on EGFR synthesis on a longer time-scale to re-establish the organization at criticality, as we have noted in the manuscript. The duration of the organization will however depend on the duration and concentration of EGF pulses, since they are directly proportional to the fraction of ligand-bound receptors that are unidirectionally removed and effectively decrease the EGFR concentration on the membrane, shifting the system in the monostable regime. Experimentally (Stanoev et al. Cell Systems 2018), for 5min EGF pulses of 20ng/ml, we indeed found that the system could maintain the transient memory and thereby the organization in the vicinity of the SN bifurcation, for two subsequent pulses.

4. The proximity of the EGFR system to its critical transition appears to have been demonstrated in a previous study (Cell Systems 2018). Therefore the purpose of the present work, beyond elaborating on the advantages of this proximity, seems to be to demonstrate its generality. However, apart from the EGFR system, the authors only appeal to a neuronal model, where transient memory and plasticity is relatively well established. Therefore, I am not convinced that the authors' have sufficiently demonstrated that their findings are as general within biochemical networks as they claim.

Indeed, in the previous manuscript we have demonstrated that the EGFR system is positioned in the vicinity of the SN bifurcation. However, in that organization, only single steady state – that of basal EGFR phosphorylation is stable, and it was therefore unclear how the system processes the information about time-varying growth factor signals. To the best of our knowledge, there is no available mechanism that describes the dynamical principles of processing non-stationary signals. This question has been generally explored for neuronal networks, where it has suggested that the computations have to occur via metastable states. However, the concept of the liquid state machines or eco-state networks (Maas et al. *Neur. Comp.* 2002, Jaeger et al., *Science* 2004) cannot be directly translated to biochemical networks, as we discuss in the paper. This is mainly because the neuronal networks are bigger and have more complex intrinsic dynamics. We have therefore identified a minimal dynamical principle – organization in a vicinity of a saddle-node and a corresponding ghost attractor that allows processing of non-stationary signals. However this principle is more basic than the ones proposed for neuronal networks and therefore it can be realized in a broad range of systems that display bistable behavior and are organized at the transition. We strongly believe that having a basic mechanism that describes processing of non-stationary signals will lead to further investigations how cell process the information about changing environments, or how intercellular networks such as genetic networks respond to time-varying signals.

Even more, in the current literature predominantly the dynamical features of bistable systems have been explored and used to explain for example switch-like dose response activation, memory etc. However, the number of studies where it has been experimentally demonstrated whether the system is organized in the bistable regime or at the border are extremely limited (Acar et al., *Nature* 2005). We show in this work that systems can acquire such optimal features as the once noted above, and additionally uniquely retain plasticity in cellular responses for organization in the vicinity of the SN bifurcation. We therefore believe that this will provide a novel insight and promote identifying where systems are organized, as it is of significant importance how systems that can display bistability process information.

Thank you again for your reply on our decision about your manuscript MSB-19-8870 entitled "Optimal biochemical information processing at criticality".

I have now had a chance to read the manuscript again as well as the referee reports and your response letter. I can see that there might have been some issues due to difficulties in understanding the text and unclarities in the presentation of some of the concepts and analyses.

I would therefore not be opposed to consider a deeply revised manuscript that would address the points of the reviewers and that would carefully consider the following editorial suggestions:

- information processing: this term appears in the title and is often used in the text, but it does not appear to be very well defined or in fact directly illustrated. For example, in Fig 1F, while the transient memory after the growth factor pulse is well depicted, it does not seem to be illustrated how this transient memory is used for actual signal processing. I would (naively?) have expected examples showing some kind of frequency-dependent signal integration based on this transient memory (eg when pulses are sufficiently close that the transient overlaps with the next pulse, etc...) or perhaps other signal integration (integral feedback?) that could crucially depend on the presence of transient memory to successfully process information to perform a given function. In Fig 3A and C, the difference between 'short-term' (reversible bistable) and 'transient' (critical) is perhaps also not as clear as it could be and the *qualitative* difference is so obvious (Fig 3C orange vs magenta).

- optimal processing: this is also a term that appears in the title and would therefore need to be clearly defined in a formal and quantitative way. The objective function used for optimality should be defined. If it is sufficient to say that the dynamical range is maximal (Fig 3E) and noise filtering is maximal (Fig 3F) then perhaps the use of 'optimality' is not essential? Also here, if there would be some downstream function that crucially depends on transient memory (as realized in the critical state), then an optimality criterion could make sense. It could be well defined and it may depend on maximal dynamical range and noise filtering properties that would not be possible in the other regimes. But this should be clearly shown and stated.

- robustness: it seems that this is essentially equivalent to 'insensitive to spurious activation'. But then 'robustness' is also mentioned in the context of Fig 3D "robust sensing via receptor activation in a switch-like manner". The exact meaning of 'robustness' should be defined and used consistently -- if it is necessary to refer to 'robustness' at all. Maybe simple plain language is clearer in the context of this study and would help to keep the focus on the key aspects of the dynamical properties reported here, which may require all the reader's attention.

- fluctuation-sensing mechanism: it seems that your study in fact reports 2 key ideas: a) the existence of a transient memory due to a 'ghost attractor' when the system is in the vicinity of the SN point; b) the proposal that a low-pass filter could couple the very property of noise filtering at the critical state to a feedback mechanism that maintains it at criticality. This second idea is very difficult to understand from the very dense description provided and is possibly important. It is not quite clear how stable this mechanism is supposed to be (eg in the case of EGFR, increased recycling rate cannot bring the system back to criticality if receptor levels fall below a given threshold; how would this be avoided? is this what happens in Fig 4D? how can it be said then that the system is 'retained' at criticality?). It remains also very unclear what properties of Akt would explain such a low-pass filter and how it would be coupled to increase recycling rate. This aspect of the study is sufficiently important that this aspect (eg Akt involvement) should be explained with actual modeling and simulations and, even better, experiments, if possible.

- generality: most of the study is framed in the context of the EGFR signaling system. This is reasonable given the focus on biochemical circuits. It would however be crucial to discuss how general this organization and dynamical properties are expected. This concerns both the critical state as the coupling with a frequency-sensing feedback mechanism. We acknowledge that there are some references to neuronal firing examples, but we feel that the discussion of biochemical circuits and potential biological functions that might benefit from the postulated dynamics are key. In this regard, we would expect to have discussion and suggestions of experimental strategies that might test the existence and relevance of such dynamics.

- accessibility of the study: the paper is now very difficult to read and as noted above the last part rather opaque. It might be due to stylistic issues. Many sentences can be simplified and perhaps the use of terms such as 'optimality', 'robustness', 'flexibility' may sometimes mask concepts that might be easier to understand if stated plainly. In terms of making the message more accessible to biologists, it might be useful to better use the section on the microscopic model reported in Fig 2 (here also, the reference to 'epidemic spreading' in the context of a molecular circuit might more confuse readers than help them?). While this section does not seem to provide fundamental insights, one idea would be to use the microscopic model to perform 2D simulations and provide intuitive 2D visualizations of the spatial dynamics in each regime (potentially also in the form of movies). Would this be possible? It might help many readers to visualize how activity spreads spatially and, in particular in the critical regime, how 'local pockets of active receptor [...] transiently sustain and further propagate this state across the surface'. This kind of visualizations might be more accessible than the current phase space diagram in Fig 2C (note: what is the meaning of the different colors).

I hope these few suggestions make sense and can help to clarify the text in addition to convincingly reply to the points raised by the reviewers. While the concerns in terms of generality and relevance of the proposed mechanism remain important, we would not be opposed to review again a deeply revised manuscript with a much clarified text and presentation and possibly further analyses that better illustrate and support the two main points made in the study - dynamics at criticality and self-organization to criticality.

We thank the referees for their constructive comments that we used as a guideline to re-structure and re-write the manuscript, and to include new results. In the amended version of the manuscript, we present more clearly our main finding: a dynamic temporal memory in receptor activity arising via a “ghost” attractor, is a pre-requisite for processing time-varying growth factor signals. We have included novel analysis to substantiate the role of a dynamic memory resulting from the metastable state for signal integration, in contrast to long-term memory resulting from a steady state attractor. Additionally, we also demonstrate that this dynamical phenomenon is different from the relaxation kinetics of receptor activity. We thus propose criticality organization as a generic mechanism for processing of time-varying growth factor signals, and use molecular details of several other receptor sensing systems to strengthen this argument. A point-by-point response is as follows:

Reviewer #1:

A large focus of dynamical systems theory is the identification of qualitative behavior of systems around steady states. Unfortunately, biological systems are often far from steady state. In this paper Stanoev et al analyze receptor-mediated signaling using EGFR as a model for benefits of non-equilibrium behavior. Overall the work is interesting, however, I had a hard time understanding the theoretical support for many of the claims made here. My recommendation is for the authors to clearly define all the different attributes they associate with the system and specifically demonstrate all their claims based on these definitions.

For example, I am having a hard time understanding why a system that has a critical point between monostable and bistable shows "plasticity and robustness". Figure 1 title is "Plasticity and robustness to non-stationary cues emerges at criticality." However, nothing in the figures specifically addresses neither plasticity or robustness. It just shows the "transient memory" that comes from the longer transient cells go through.

In Figure 1 (now Figure 2) we have demonstrated the dynamical principle how cell surface receptors can maintain memory in receptor activity even after growth factor signal has been removed. The criticality organization confers switch-like activation to threshold growth factor dose, thus a noise-filtering, and thereby robust signaling response (current Figure 1D), but on the other hand the activation is also reversible – enabling plasticity in responsiveness to time-varying growth factor signals (current Figure 3A,B). As demonstrated, these two features uniquely emerge at the critical transition. However, we agree with the referee that this terminology was not accompanied with sufficient explanations to support clarity, and therefore, in the amended version of the manuscript we have rephrased our statements.

We would like to emphasize here that the prolonged receptor activity upon signal removal – the transient memory, is not a simple slow relaxation,

consequence of the kinetic parameters of the system, but results from the closeness to the saddle-node bifurcation, and thereby has completely different features than typical transient behavior. The state trajectory is transiently maintained at the “ghost” position, encoding the memory of previous activity. To demonstrate this, we analyzed the receptor activity response to different growth factor pulse trains and calculated the number of disjoint intervals of receptor activity over the integration time. The respective distributions for 1000 independent realizations, when the system is positioned in the different parameter regimes is shown in newly included Fig.3C (the details of the simulations are given in the manuscript). The distribution is broad only for the critical organization, demonstrating that the dynamic memory enables signal integration of varying degrees, adapting to the temporal signatures of the pulse trains (representative examples of temporal receptor activity profiles for different growth factor pulse trains are given in Fig.3D). In contrast, for monostable organization, the skewed distribution is narrow and illustrates that signal integration generally does not occur, except in rare cases where adjacent pulses are in very close proximity, smaller than the rapid, kinetics-driven relaxation time to basal activity level.

Similarly, the author claims about the special positioning of critical point that: "In this organization, the system does not process information using the stable attractors associated with specific states (0-1 case), but rather the information is maintained via the transient memory and interpreted through the phase-space trajectories". I really don't understand what kind information processing is happening there... this is just a long transient behavior.

Under information processing we refer here to receptor activity response in time that is not a simple steady-state output reflection of the immediate input signal, but has an additional temporal component that encodes and is dependent on the history of signal changes. In the amended version of the manuscript, we have substantiated this statement by demonstrating the receptor's response to complex temporal growth factor pulse trains for systems organization in the different dynamical regimes (monostable, reversible bistable, criticality, irreversible bistable, new Fig.3). Regarding the “transient behavior”, as noted in the response to the previous question, this is not a long (exponential) decay to basal state but the system occupies an intermediate, metastable state (Fig.2C). In the current version of the manuscript, we discuss more clearly how this transient memory differs from the long-term memory that results from organization in the bistable regime, where the two stable states correspond to basal receptor phosphorylation level (0 level) and high receptor phosphorylation level (level 1).

In figure 3 the author claim that the system shows "Signal integration and optimal information processing". I don't understand what is the support for these claims. What is the optimality criteria used? What signals are being integrated? Why is this unique to systems that pass through a critical point?

We thank the referee for pointing to us where clarity in the presentation of our findings was missing. As already noted in the responses above, to

demonstrate the uniqueness of the metastable state (generated for critical organization) for signal integration, we generated 1000 independent growth factor pulse trains. The realizations were used to quantify the temporal receptor's activity response through the total duration of receptor activity and the number of disjoint interval of receptor activity for each of the different pulse realizations (new Fig.3B-D). We show that only in the critical organization, the total duration of receptor activity depends on the temporal growth factor profile.

We also avoid to use qualifiers, but only present the features of the systems from the distinct dynamical regimes and discuss which features are compatible with sensing and interpreting time-varying growth factor signals.

Finally, the speculative model in figure 4 that suggests that it would be good for the system to sense fluctuation in concentration is 1) not supported by any data 2) not clear to me why it is better than direct sensing of concentration. Why is it less sensitive to noise?

In Fig.4 we proposed a mechanism how the system can maintain itself exactly at this critical point between the mono- and bistable mode of operation in the face of parameter perturbations, namely depletion of receptors from the membrane. We are not aware of any example in the literature discussing a mechanism where cells can sense and thereby dynamically regulate receptor concentration on the plasma membrane in a specific interval. We therefore propose that receptor activity fluctuation sensing via a low-pass filter is one possible mechanism for concentration sensing that by coupling to receptor recycling would maintain positioning of the system in the vicinity of the saddle-node bifurcation. This mechanism is probable since this adaptive feedback has the highest sensitivity to be triggered in this region.

This represents a possible hypothesis for the EGFR system that we base on previous experimental findings: that the EGFR system sits on the transition between mono- and bistable mode of operation, and that receptor concentration perturbations affect the temporal responses (Stanoev et al., Cell Systems, 2018), that the sensing and actuation mechanism could be possibly realized via the network motifs in which Akt is embedded since Akt promotes EGFR recycling (Stallaert et al., Sci. Signaling 2018) and Akt has been suggested to be embedded in a low-pass filter (Fujita et al., Sci. Signaling 2010).

In the current version of the manuscript, we clarify more explicitly that this is a proposed hypothesis.

Minor points:

- 1. Figure 1 is very hard to follow with all the arrows, potentials, etc. I recommend simplifying or revising it.*

In the amended version of the manuscript (current Figure 2), we have re-organized the figure to make the arrows more compact in order to represent the subsequent changes in the geometry/topology of the phase space that guide the receptor's responses for the different organizations. We also reorganized the figures for clarity.

- 2. To emphasize the generalities of their findings and universality of the approach the authors analyzed the receptor system using an epidemic model and also include a model of a leaky neuron. In my opinion (and I understand that this could reflect personal preference) these are distracting and should be removed from the paper.*

In order to quantify the spreading and maintenance of receptor activity over cell membrane on the molecular level, we used the basic reproduction number (R_0), introduced for epidemic spreading in networks. Given that we had also analytically derived R_0 for the receptor system used, in the amended version of the manuscript we restrain from the analogy to epidemic spreading, but use R_0 to explain the transmission potential for the distinct dynamical regimes (we have however maintained the relevant literature references for the definition of R_0).

Following the suggestion of the referee, we have removed the neuronal model.

- 3. The authors use a lot of dynamical systems jargon. It will be more accessible to broader readership if they try to reduce the use of such jargon.*

Upon the referee's suggestion, we have tried to amend the language where possible. We decided however to retain part of the terminology we believed was necessary to describe the unique dynamical features that emerge for systems organization in a vicinity of a saddle-node bifurcation.

Reviewer #2:

The manuscript by Stanoev et al investigates the properties of an input-output response near a dynamical transition between a monostable and a bistable state. The authors focus on a toggle-switch-type signaling network inspired by the EGFR network. Their main result is that, after switching from the low to high state, the transition back to the low state (i) is fast if the high state disappears, or (ii) may never occur if the high state is retained, but (iii) can be slow following a transient if the high state is "almost" retained (the so-called "ghost attractor) which occurs near the dynamical critical point. They show that this behavior persists in a spatially resolved version of their receptor signaling model, as well as in an unrelated neuronal model, and they make an argument for how the EGFR network might keep itself near this critical point using self-organization (i.e. corrective feedback).

Understanding general principles of memory and plasticity in signaling networks is an important goal that is highly relevant to the field of molecular systems biology. However, there are concerns with this manuscript that make it unclear whether it reaches this goal. The concept of a ghost attractor is

intuitive and well studied.

Although the concept of a ghost attractor is known (Strogatz and Westervelt, PRB, 1989), to the best of our knowledge, it has not been proposed as a possible mechanism that can generate dynamic memory in sensing systems. In the amended version of the manuscript, we elaborate in a greater extent (and include new simulations, new Figure 3) that the thereby generated metastable state is a unique dynamical mechanism that enables receptor activity in time to reflect the history of the temporal growth factor signal profile. In contrast, the current models of multistability and switching between stable attractors, as we demonstrate in the manuscript, are limited in this regard. We also elaborate in the introduction on examples where dynamic memory is crucial.

Moreover, the concept of a ghost attractor has been somewhat investigated in neuronal networks (Izhikevich, 2007), however, even there, the possibility of a ghost attractor to serve as a dynamical mechanism for dynamic memory that can serve to integrate multiple signals has not been proposed. The current concepts of dynamic memory in the literature rely on chaotic, feed-forward or attractor networks, and their limitations in the literature are clearly identified (Murray et al., PNAS, 2017). Although we have opted to exclude the neuronal examples from the amended version of the manuscript for clarity, we argue why that the proposed dynamical mechanism for processing time-varying signals is novel and brings a basic principle how cells process information from non-stationary environments.

Therefore, the novelty of this study would potentially be to explain the mechanism more deeply and/or demonstrate that it occurs across a wide range of biochemical systems. I am not sure that it sufficiently achieves either of these goals.

The principles of processing time-varying signals with dynamic memory as proposed in our manuscript relies only on the closeness to a saddle-node bifurcation. In the amended version of the manuscript we included a more detailed analysis for signals with complex temporal patterns and compare the criticality with the remaining dynamical regimes. This mechanism will therefore apply to any biochemical system that is organized in the vicinity of the saddle-node bifurcation. Due to the large number of biochemical networks that exhibit bistable behavior (e.g. various receptor networks, the MAPK network, gene regulatory networks during development etc.) the potential to expand this study with multiple other examples that are also governed by different type of dynamics (i.e. Hill-type dynamics) is very large. Since we have restricted our analysis to receptor systems, in the amended version of the manuscript we included a more in-depth discussion regarding other receptor systems that have similar molecular regulation mechanism (other members of the RTK family, G-protein coupled receptors), and for which we expect the same principle to hold true.

Specifically:

1. It is expected that both sensitivity and noise are maximal near a critical transition. However, the authors find that sensitivity is maximal (Fig 3E) but noise is maximal on one side and not the other (Fig 3F). What is the mechanism for this? Currently it is unclear because the results are purely numerical.

We apologize that these points have not been clearly explained in the manuscript - in the previous Figure 3F we have shown the propensity for spontaneous activation, and not the amplitude of the noise.

In contrast to critical phenomena characterizing phase transitions, the features of the system in the critical vicinity of different bifurcations such as saddle-node or pitchfork bifurcation are different. It has been experimentally demonstrated for example for an external-cavity laser diode (Rogister et al., PRE 2003) that the frequency of the fluctuations is minimal in the vicinity of the transition due to the saddle-node ghost, a principle that we also use and demonstrate in Fig.4. This feature is only characteristic for a saddle-node bifurcation. In the amended version of the manuscript however, we decided not to present former Fig.3F for clarity.

2. Relatedly, in the EGFR model the authors claim that the noise is lowest near the transition (Fig 4E). However, the plot of noise vs the bifurcation parameter (blue curve in Fig 4E) is wildly fluctuating, and I am not convinced that it has a clear minimum. This underscores the confusion about the role of noise near the transition.

As noted in the answer to the previous comment, we demonstrate that the frequency of the fluctuations rather than the noise amplitude itself has a minimum in the vicinity of the transition due to the saddle-node ghost. This is a characteristic behavior in a vicinity of a SN bifurcation based on which we postulate that cells can sense parameter space organization (EGFR concentration) by using a low-pass filter, and subsequently maintain positioning in the SN vicinity.

3. The authors aim to present a plausible mechanism for the EGFR system to retain itself near the ghost attractor (Fig 4). However, the mechanism does not appear to succeed after the second pulse (Fig 4D). This makes it hard to understand whether their self-organization argument applies.

Indeed, the mechanism can only provide a short-term maintenance at the SN bifurcation by tipping the plasma membrane vs endosomal receptor concentration balance to counteract the receptor removal from the membrane. This is dynamically limited, since the SN positioning asymptotically approaches a receptor concentration below which the recycling rate cannot compensate for the loss of receptors on the membrane, as demonstrated by the bifurcation analysis (current Fig. 5C). Additionally, the receptor recycling machinery might also impose an upper bound on the recycling rate, further limiting the resetting capacity. The cell will therefore rely on EGFR synthesis on a longer time-scale to re-establish the organization at criticality, as we have noted in the manuscript. The duration of the organization will however depend on the duration and amplitude of EGF pulses, since they directly

translate to the fraction of ligand-bound receptors that are unidirectionally removed and effectively decrease the EGFR concentration on the membrane, shifting the system towards the monostable regime. Experimentally (Stanoev et al. Cell Systems 2018), for 5min EGF pulses of 20ng/ml, we indeed found that the system could maintain the transient memory and thereby the organization in the vicinity of the SN bifurcation, for two subsequent pulses.

4. The proximity of the EGFR system to its critical transition appears to have been demonstrated in a previous study (Cell Systems 2018). Therefore the purpose of the present work, beyond elaborating on the advantages of this proximity, seems to be to demonstrate its generality. However, apart from the EGFR system, the authors only appeal to a neuronal model, where transient memory and plasticity is relatively well established. Therefore, I am not convinced that the authors' have sufficiently demonstrated that their findings are as general within biochemical networks as they claim.

Indeed, in the previous manuscript we have demonstrated that the EGFR system is positioned in the vicinity of the SN bifurcation. However, in that organization, only single steady state – that of basal EGFR phosphorylation is stable, and it was therefore unclear how the system processes the information about time-varying growth factor signals. To the best of our knowledge, there is no available mechanism that describes the dynamical principles of processing non-stationary signals for biochemical networks. This question has been generally explored for neuronal networks, where it has been suggested that the computations have to occur via state-dependent trajectories. However, the concept of the liquid state machines or eco-state networks (Maas et al. Neur. Comp. 2002, Jaeger et al., Science 2004) cannot be directly translated to biochemical networks, as we discuss in the paper. This is mainly because the large-scale neuronal networks have more complex intrinsic dynamics. We have therefore identified a minimal dynamical principle – organization in a vicinity of a saddle-node and a corresponding ghost attractor that allows processing of non-stationary signals. However this principle is more basic than the ones proposed for neuronal networks and therefore it can be realized in a broad range of systems that display bistable behavior and are organized at the transition. We strongly believe that having a basic dynamical mechanism that describes processing of non-stationary signals will increase further experimental investigations how cell process the information about changing environments, or how intercellular networks such as genetic networks respond to time-varying signals.

Thank you again for submitting your work to Molecular Systems Biology. We have now heard back from referee #2 who accepted to evaluate the study. As you will see, the reviewer is now cautiously supportive but still raises some issues. In particular the point that a similar behavior is "expected to be observed for an long exponential decay with the same effective timescale as the long non-exponential decay of the ghost attractor" should be addressed.

REFEREE REPORTS

Reviewer #2:

The authors have made several improvements to the manuscript. Specifically, they have clarified the ghost attractor mechanism using new diagrams and simulations in Figs 1 and 3, and they have removed the neuroscience example while expanding a discussion of their results for receptor networks more generally.

There are still shortcomings and unresolved reviewer questions. Specifically, it is unclear why the main result is unique to systems that pass through a critical point (Reviewer 1), given that the same "stay high for repeated pulses" behavior is expected to be observed for an long exponential decay with the same effective timescale as the long non-exponential decay of the ghost attractor. Similarly, the shortcomings of the self-organization analysis remain (Reviewer 2): the minimum in the noise (whether amplitude or frequency) shown now as the blue curve in Fig 5E is unconvincing due to the wild fluctuations and the < 2% range of the curve; and the fact that the adaption is only maintained for two pulses (Fig 5D) is now better elucidated but still undercuts the utility of the mechanism.

Essentially, the study boils down to the utility of the ghost attractor for pulse integration. This is now well described in Fig 1-3. Fig 4 and 5 demonstrate extensions using space and adaptation, but the main point is the same.

Is it enough for Molecular Systems Biology? Frankly, the results are not super surprising and therefore it reads like a relatively simple idea that gets oversold. But the mechanisms are well described, the execution is good, and efforts are made to take the work in realistic directions. I think this could be fine for MSB, but ultimately it is an editorial call.

We thank the referee for the additional comments on the manuscript, which we addressed as follows:

Reviewer #2:

The authors have made several improvements to the manuscript. Specifically, they have clarified the ghost attractor mechanism using new diagrams and simulations in Figs 1 and 3, and they have removed the neuroscience example while expanding a discussion of their results for receptor networks more generally.

There are still shortcomings and unresolved reviewer questions. Specifically, it is unclear why the main result is unique to systems that pass through a critical point (Reviewer 1), given that the same "stay high for repeated pulses" behavior is expected to be observed for an long exponential decay with the same effective timescale as the long non-exponential decay of the ghost attractor.

In the previous version of the manuscript, we have addressed this point by demonstrating the signal integration capabilities for diverse organization of the system in parameter space. In that sense, for organization in the monostable regime, the degree of history-dependent signal integration only depends on the kinetic parameters of the system. We note once again that a transient memory in receptor activity that results from a metastable state is characterized by transient maintenance of steady levels of high receptor activity before rapidly declining to basal activity levels. This is clearly distinct from an exponentially decaying relaxation process that continuously declines. In order to address this question more explicitly however, we have now modeled a receptor activity profile according to

$$\frac{dR_a}{dt} = I * \alpha * (R_T - R_a) - \beta * R_a \quad \text{Eq. (1)}$$

where I denotes the growth factor pulse, α – the activation and β - the decay rate constant, whereas R_T reflects the total receptor concentration. We fitted this equation to the numerical receptor activity profile obtained upon single pulse stimulation for organization at criticality (Fig.2c in the manuscript), in order to match the time scale of the exponential decay of Eq. (1) (light red, Fig.1a) to the time-scale of the non-exponential decay determined by the presence of the “ghost” attractor (magenta, Fig.1a). In addition, we also fitted a slower exponential decay (dark red, Fig.1a) to the response profile at criticality, such that their response amplitudes are matched at the ending time point of the transient memory.

Figure 1. Comparison between information processing features emerging from a transient dynamic memory and an exponentially decaying relaxation process. *a.* Receptor response profile upon single short input pulse (yellow shaded region) for critical organization of Ra-P_{DNF} system (magenta), in contrast to response profile following Eq. (1) for $\beta = 5.5 \times 10^{-4}$ (dark red) and $\beta = 0.035$ (light red). *b.* Respective response profiles upon train of growth factor pulses. Corresponding degree of *c.* history-dependent duration of receptor activity and *d.* partitioning in response to complex signals for both cases. The distributions are estimated from 1000 different realizations of growth factor pulse trains as in Fig.3 in the manuscript. Other parameters: $\alpha = 0.023$ and $R_T = 0.8351$ (dark red) and 0.9416 (light red). Parameters for the magenta profile as in Fig.2c of the manuscript.

The simulations depicting receptor response upon a growth factor pulse train demonstrated that in the first case, when the time scale of the exponential decay of Eq. (1) matches the time-scale of the non-exponential decay determined by the presence of the “ghost” attractor, the total duration of high receptor activity (\geq transient memory amplitude) is proportional to the total duration of growth factor pulses (light red, Fig.1c). This indicates a low degree of signal integration, resembling the behavior of the R-P_{DNF} system for monostable organization. On the other hand, when slower exponential decay was considered (dark red, Fig.1b), the variance in the total duration of high receptor activity resembles the one estimated for the critical R-P_{DNF} organization (Fig.1c, dark red and manuscript Fig.3b, magenta). However in this case, the number of disjoint intervals of receptor activity over the integration time differs. In particular, due to the long exponential decay, the system cannot reset to basal activity state, as reflected through the narrow distribution of the number of disjoint intervals close to 1 (dark red, Fig.1d). Thus, a simple exponential decay relaxation process cannot capture the

information processing capabilities of the transient dynamic memory emerging from a “ghost” attractor, where high activity state is maintained transiently, followed by a rapid relaxation to basal levels. For a relaxation process, there is trade-off between the degree of history-dependent signal integration and the degree of partitioning in response to the complex signal (compare distributions in Figs.1c and d). In the amended version of the manuscript, we have included an equivalent analysis for five different decay rates (Fig. EV3) and also discuss the respective findings.

Similarly, the shortcomings of the self-organization analysis remain (Reviewer 2): the minimum in the noise (whether amplitude or frequency) shown now as the blue curve in Fig 5E is unconvincing due to the wild fluctuations and the < 2% range of the curve; and the fact that the adaption is only maintained for two pulses (Fig 5D) is now better elucidated but still undercuts the utility of the mechanism.

We would like to note that Fig.5 only represents a conceptual model of a possible mechanism how organization at the critical transition can be maintained. As a reminder, it relies on the generic observation that minimal frequency of fluctuations is observed in the vicinity of a SN bifurcation, as demonstrated previously for technical systems (referenced in the manuscript, along with an explanation of this effect). Thereby, the Figure displays qualitative rather than quantitative properties of a possible organization mechanism. The number of pulses to which the cell will be sensitive, as described in the manuscript, depends on the duration of the pulses as well as the growth factor dose. Since our model is scaled to match the experimental observations of the EGFR sensing system (Stanoev et al., 2018), the memory will be maintained for 2 subsequent 5min pulses of 20ng/ml EGF, that are 30min apart, as experimentally demonstrated. The trafficking of the EGFR receptor species, as for other RTKs, that promote uni-directional removal of ligand-bound receptors from the plasma membrane would however impose a bound on any dynamical resetting mechanism, thus requiring synthesis of new receptors to recover the critical organization of the system.

Essentially, the study boils down to the utility of the ghost attractor for pulse integration. This is now well described in Fig 1-3. Fig 4 and 5 demonstrate extensions using space and adaptation, but the main point is the same.

Is it enough for Molecular Systems Biology? Frankly, the results are not super surprising and therefore it reads like a relatively simple idea that gets oversold.

Respectfully, we strongly disagree with the referee that the results are “not surprising” and that we present a “relative simple idea that is oversold”. The question of the dynamical mechanism that enables cells to integrate time-varying growth factor signals, to the best of our knowledge, has not been addressed in the literature, neither it has been described that memory in receptor activity is necessary for signal integration. We identified a minimal dynamical mechanism that underlies both of these features in receptor networks. Information processing capabilities arising from a metastable state

cannot be achieved by a simple exponentially decaying relaxation process, making the proposed mechanism not trivial (Response to point 1 and Figure EV3 in the manuscript). This we substantiate through the fact that metastable state uniquely provides history-dependent duration of receptor activity as well as adaptation of the response to the temporal partitioning of the signal. For an exponentially decaying process, either one or the other feature is lost. The manifestation of the metastable state that transiently maintains steady levels of high receptor activity before quickly resetting to basal levels is thereby crucial for processing time-varying signals.

We would like to underline once more the fact that dynamical basis of processing time-varying signals has been only explored in neuronal networks so far, where a similar concept - processing with transient trajectories through phase-space has been proposed, as elaborated by the “echo state networks” or “liquid-state machines” (Maas et al., 2002; Jaeger, 2001). However, what is distinct in our work is that we propose a “ghost” attractor as a minimal dynamical mechanism enabling signal integration features even by biochemical networks, whose dynamics is much more restricted than the one of neuronal networks.

But the mechanisms are well described, the execution is good, and efforts are made to take the work in realistic directions. I think this could be fine for MSB, but ultimately it is an editorial call.

Accepted

23rd December 2011

Thank you again for sending us your revised manuscript. We are now satisfied with the modifications made and I am pleased to inform you that your paper has been accepted for publication.

USEFUL LINKS FOR COMPLETING THIS FORM

<http://www.antibodypedia.com>
<http://1degreebio.org>
<http://www.equator-network.org/reporting-guidelines/improving-bioscience-research-repor>

<http://grants.nih.gov/grants/olaw/olaw.htm>
<http://www.mrc.ac.uk/Ourresearch/Ethicsresearchguidance/Useofanimals/index.htm>
<http://ClinicalTrials.gov>
<http://www.consort-statement.org>
<http://www.consort-statement.org/checklists/view/32-consort/66-title>

<http://www.equator-network.org/reporting-guidelines/reporting-recommendations-for-tum>
<http://datadryad.org>

<http://figshare.com>

<http://www.ncbi.nlm.nih.gov/gap>

<http://www.ebi.ac.uk/ega>

<http://biomodels.net/>

<http://biomodels.net/miriam/>
<http://jil.biochem.sun.ac.za>
http://oba.od.nih.gov/biosecurity/biosecurity_documents.html
<http://www.selectagents.gov/>

Corresponding Author Name: Aneta Koseska
 Journal Submitted to: Molecular Systems Biology
 Manuscript Number: MSB-19-8870